# DUAL ASSOCIATED ENCODER FOR FACE RESTORATION

**Yu-Ju Tsai**[1]    **Yu-Lun Liu**[2]    **Lu Qi**[1*]   **Kelvin C.K. Chan**[3]    **Ming-Hsuan Yang**[1,3]

[1]UC Merced    [2]National Yang Ming Chiao Tung University    [3]Google Research

## ABSTRACT

Restoring facial details from low-quality (LQ) images has remained challenging due to the nature of the problem caused by various degradations in the wild. The codebook prior has been proposed to address the ill-posed problems by leveraging an autoencoder and learned codebook of high-quality (HQ) features, achieving remarkable quality. However, existing approaches in this paradigm frequently depend on a single encoder pre-trained on HQ data for restoring HQ images, disregarding the domain gap and distinct feature representations between LQ and HQ images. As a result, encoding LQ inputs with the same encoder could be insufficient, resulting in imprecise feature representation and leading to suboptimal performance. To tackle this problem, we propose a novel dual-branch framework named *DAEFR*. Our method introduces an auxiliary LQ branch that extracts domain-specific information from the LQ inputs. Additionally, we incorporate association training to promote effective synergy between the two branches, enhancing code prediction and restoration quality. We evaluate the effectiveness of DAEFR on both synthetic and real-world datasets, demonstrating its superior performance in restoring facial details. Project page: https://liagm.github.io/DAEFR/.

## 1 INTRODUCTION

Blind face restoration presents a formidable challenge as it entails restoring facial images that are degraded by complex and unknown sources of degradation. This degradation process often results in the loss of valuable information. It introduces a significant domain gap, making it arduous to restore the facial image to its original quality with high accuracy. The task itself is inherently ill-posed, and prior works rely on leveraging different priors to enhance the performance of restoration algorithms. Notably, the codebook prior emerges as a promising solution, showcasing its effectiveness in generating satisfactory results within such challenging scenarios. By incorporating a codebook prior, these approaches demonstrate improved performance in blind face restoration tasks.

Existing codebook methods (Zhou et al., 2022; Gu et al., 2022; Wang et al., 2022b) address the inclusion of low-quality (LQ) images by adjusting the encoder, which is pre-trained on high-quality (HQ) data, as depicted in Fig. 1(a). However, this approach introduces domain bias due to a domain gap and overlooks the distinct feature representations between the encoder and LQ images. Consequently, employing the pre-trained encoder to encode LQ information may potentially result in imprecise feature representation and lead to suboptimal performance in the restoration process. Furthermore, these approaches neglect the LQ domain's inherent visual characteristics and statistical properties, which provide valuable information for enhancing the restoration process.

To overcome these limitations, we present a novel framework called DAEFR, which incorporates a dedicated auxiliary branch for LQ information encoding (Fig. 1(b)). This auxiliary branch is exclusively trained on LQ data, alleviating domain bias and acquiring a precise feature representation of the LQ domain. By integrating the auxiliary branch into our framework, we effectively harness the enhanced LQ representation of identity and content information from the original LQ images, which can supplement the lost information. DAEFR utilizes both HQ and auxiliary LQ encoders to capture domain-specific information, thereby enhancing the representation of image content.

The core idea of our method is to effectively fuse visual information from two branches. If we naively combine the HQ and LQ features, the existing domain gap causes misalignment in feature

---

*Corresponding author

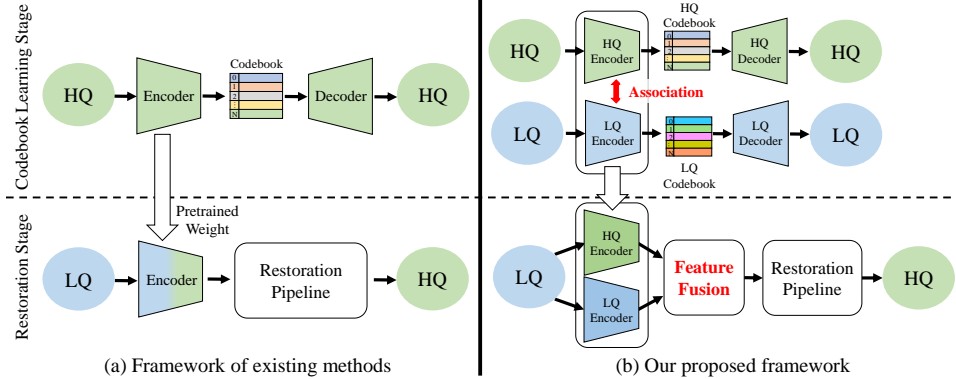

Figure 1: **Comparison to existing framework.** (a) Existing codebook prior approaches learn an encoder in the first stage. During the restoration stage, these approaches utilize LQ images to fine-tune the encoder using pre-trained weights obtained from HQ images. However, this approach introduces a domain bias due to a domain gap and overlooks the distinct feature representations between the encoder and LQ input images. (b) In the codebook learning stage, we propose the integration of an auxiliary branch specifically designed for encoding LQ information. This auxiliary branch is trained exclusively using LQ data to address domain bias and obtain precise feature representation. Furthermore, we introduce an association stage and feature fusion module to enhance the integration of information from both encoders and assist our restoration pipeline.

representation, rendering them challenging to utilize effectively. Similar to CLIP (Radford et al., 2021), we incorporate an association stage after extracting features from both branches. This stage aligns the features to a shared domain, effectively bridging the gap between LQ and HQ features and facilitating a more comprehensive and integrated representation of feature information. To ensure the effective fusion of information, we employ a multi-head cross-attention module after acquiring the associated encoders that can adequately represent both the HQ and LQ domains. This module enables us to merge the features from these associated encoders and generate fused features. Through the fusion process, where the features from the associated encoders are combined with the LQ domain information, our approach effectively mitigates the challenges of domain gap and information loss and leverages the complementary aspects of the HQ and LQ domains, leading to improved restoration results. The main contributions of this work are:

- We introduce an auxiliary LQ encoder to construct a more precise feature representation that adeptly captures the unique visual characteristics and statistical properties inherent to the LQ domain.
- By incorporating information from a hybrid domain, our association and feature fusion methods effectively use the representation from the LQ domain and address the challenge of domain gap and information loss in image restoration, resulting in enhanced outcomes.
- We propose a novel approach, DAEFR, to address the challenging face restoration problem under severe degradation. We evaluate our method with extensive experiments and ablation studies and demonstrate its effectiveness with superior quantitative and qualitative performances.

## 2 RELATED WORK

**Face Restoration.**    Blind face restoration techniques commonly exploit the structured characteristics of facial features and incorporate geometric priors to achieve desirable outcomes. Numerous methods propose the use of facial landmarks (Chen et al., 2018; Kim et al., 2019; Ma et al., 2020; Zhang & Wu, 2022), face parsing maps (Chen et al., 2021; Shen et al., 2018; Yang et al., 2020), facial component heatmaps (Wang et al., 2019; Chen et al., 2021; 2020; Kalarot et al., 2020), or 3D shapes (Hu et al., 2020; Ren et al., 2019; Zhu et al., 2022; Hu et al., 2021). However, accurately acquiring prior information from degraded faces poses a significant challenge, and relying solely on geometric priors may not yield adequate details for HQ face restoration.

Various reference-based methods have been developed to address these limitations (Dogan et al., 2019; Li et al., 2020; 2018). These methods typically rely on having reference images that share the same identity as the degraded input face. However, obtaining such reference images is often impractical or not readily available. Other approaches, such as DFDNet (Li et al., 2020), construct HQ facial component features dictionaries. However, these component-specific dictionaries may lack the necessary information to restore certain facial regions, such as skin and hair.

To address this issue, approaches utilize generative facial priors from pre-trained generators, such as StyleGAN2 (Karras et al., 2019). These priors are employed through iterative latent optimization for GAN inversion (Gu et al., 2020; Menon et al., 2020) or direct latent encoding of degraded faces (Richardson et al., 2021). However, preserving high fidelity in the restored faces becomes challenging when projecting degraded faces into the continuous infinite latent space.

On the other hand, GLEAN (Chan et al., 2021; 2022), GPEN (Yang et al., 2021), GFPGAN (Wang et al., 2021), GCFSR (He et al., 2022), and Panini-Net (Wang et al., 2022a) incorporate generative priors into encoder-decoder models, utilizing additional structural information from input images as guidance. Although these methods improve fidelity, they heavily rely on the guidance of the inputs, which can introduce artifacts when the images are severely corrupted.

Most recently, diffusion models (Sohl-Dickstein et al., 2015; Ho et al., 2020) have been developed for generating HQ content. Several approaches (Yue & Loy, 2022; Wang et al., 2023) exploit the effectiveness of the diffusion prior to restore LQ face images. However, these methods do not preserve the identity information present in the LQ images well.

**Vector Quantized Codebook Prior.** A vector-quantized codebook is introduced in the VQ-VAE framework (Van Den Oord et al., 2017). Unlike continuous outputs, the encoder network in this approach generates discrete outputs, and the codebook prior is learned rather than static. Subsequent research proposes various enhancements to codebook learning. VQVAE2 (Razavi et al., 2019) introduces a multi-scale codebook to improve image generation capabilities. On the other hand, VQGAN (Esser et al., 2021) trains the codebook using an adversarial objective, enabling the codebook to achieve high perceptual quality.

Recently, codebook-based methods (Wang et al., 2022b; Gu et al., 2022; Zhou et al., 2022; Zhao et al., 2022) explore the use of learned HQ dictionaries or codebooks that contain more generic and detailed information for face restoration. CodeFormer (Zhou et al., 2022) employs a transformer to establish the appropriate mapping between LQ features and code indices. Subsequently, it uses the code index to retrieve the corresponding feature in the codebook for image restoration. RestoreFormer (Wang et al., 2022b) and VQFR (Gu et al., 2022) attempt to directly incorporate LQ information with the codebook information based on the codebook prior. However, these methods may encounter severe degradation limitations, as the LQ information can negatively impact the HQ information derived from the codebook.

## 3 METHOD

Our primary objective is to mitigate the domain gap and information loss that emerge while restoring HQ images from LQ images. This challenge has a substantial impact on the accuracy and effectiveness of the restoration process. To address this issue, we propose an innovative framework with an auxiliary LQ encoder incorporating domain-specific information from the LQ domain. Furthermore, we utilize feature association techniques between the HQ and LQ encoders to enhance restoration.

In our framework, we first create discrete codebooks for both the HQ and LQ domains and utilize vector quantization (Esser et al., 2021; Van Den Oord et al., 2017) to train a quantized autoencoder via self-reconstruction (Sec. 3.1). Then, we introduce a feature association technique in a way similar to the CLIP model (Radford et al., 2021) to associate two encoders and aim to reduce the domain gap between the two domains (Sec. 3.2). In the next stage, a feature fusion module is trained using a multi-head cross-attention (MHCA) technique to combine the features extracted from the two associated encoders ($E_H^A$ and $E_L^A$). We employ the transformer to perform the code prediction process using the integrated information from both encoders, which predicts the relevant code elements in the HQ codebook. Subsequently, the decoder utilizes the restored code features to generate HQ images (Sec. 3.3). Our framework is illustrated in Fig. 2.

### 3.1 DISCRETE CODEBOOK LEARNING STAGE

Similar to VQGAN (Esser et al., 2021), our approach involves encoding domain-specific information through an autoencoder and codebook, enabling the capture of domain characteristics during the training phase. Both HQ and LQ paths are trained using the same settings to ensure consistency in feature representation. Here, we present the HQ reconstruction path as an illustrative example, noting that the LQ reconstruction path follows an identical procedure.

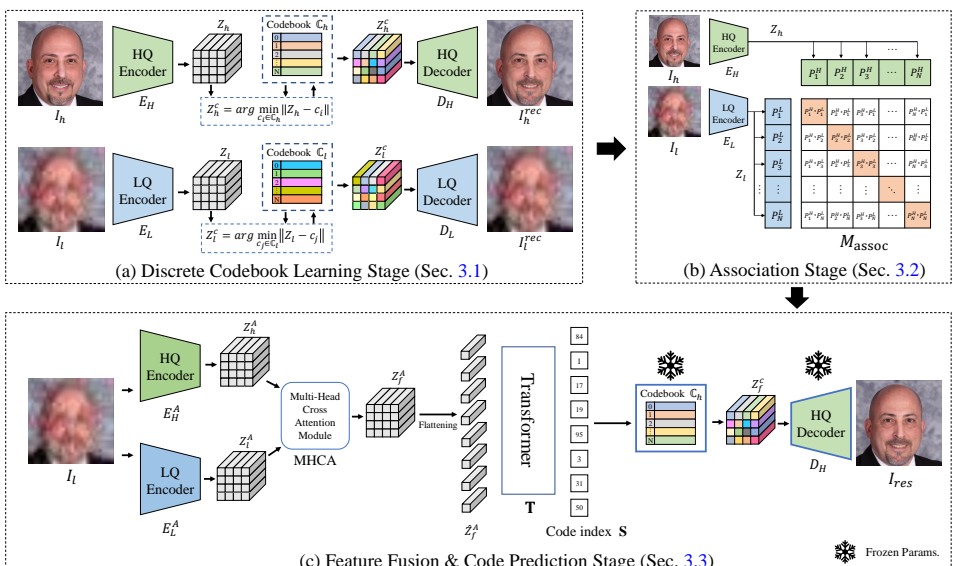

Figure 2: **Proposed DAEFR framework.** (a) Initially, we train the autoencoder and discrete codebook for both HQ and LQ image domains through self-reconstruction. (b) Once we obtain both encoders ($E_H$ and $E_L$), we divide the feature ($Z_h$ and $Z_l$) into patches ($P_i^H$ and $P_i^L$) and construct a similarity matrix $M_{\text{assoc}}$ that associates HQ and LQ features while incorporating spatial information. To promote maximum similarity between patch features, we employ a cross-entropy loss function to maximize the diagonal of the matrix. (c) After obtaining the associated encoders ($E_H^A$ and $E_L^A$), we use a multi-head cross-attention module (MHCA) to merge the features ($Z_h^A$ and $Z_l^A$) from the associated encoders, generating fused features $Z_f^A$. We then input the fused feature $Z_f^A$ to the transformer **T**, which predicts the corresponding code index **s** for the HQ codebook $\mathbb{C}_h$. Finally, we use the predicted code index to retrieve the features and feed them to the HQ decoder $D_H$ to restore the image.

The process shown in Fig. 2(a) involves the encoding of the HQ face image $I_h \in \mathbb{R}^{H \times W \times 3}$ into a compressed feature $Z_h \in \mathbb{R}^{m \times n \times d}$ by the encoder $E_H$. This step is carried out by replacing each feature vector of $Z_h$ with the nearest code item in the learnable codebook $\mathbb{C}_h = c_k \in \mathbb{R}^{d}{}_{k=0}^{N-1}$. Consequently, we obtain the quantized feature $Z_h^c \in \mathbb{R}^{m \times n \times d}$:

$$Z_h^{c(i,j)} = \arg \min_{c_k \in \mathbb{C}_h} \|Z_h^{(i,j)} - c_k\|_2, \quad k \in [0, ..., N-1], \tag{1}$$

where $Z_h^{c(i,j)}$ and $Z_h^{(i,j)}$ are the feature vectors on the position $(i, j)$ of $Z_h^c$ and $Z_h$. $\| \cdot \|_2$ is the L2-norm. After obtaining the quantized feature $Z_h^c$, the decoder $D_H$ then proceeds to reconstruct the HQ face image $I_h^{\text{rec}} \in \mathbb{R}^{H \times W \times 3}$ using $Z_h^c$.

**Training Objectives.** Similar to the prior arts (Gu et al., 2022; Zhou et al., 2022; Wang et al., 2022b; Esser et al., 2021) setting, We incorporate four distinct losses for training, which includes three image-level reconstruction losses (i.e., L1 loss $\mathcal{L}_1$, perceptual loss $\mathcal{L}_{\text{per}}$ (Zhang et al., 2018; Johnson et al., 2016), and adversarial loss $\mathcal{L}_{\text{adv}}$ (Isola et al., 2017)) and one code-level loss $\mathcal{L}_{\text{code}}$ (Esser et al., 2021; Van Den Oord et al., 2017):

$$\mathcal{L}_1 = \|I_h - I_h^{\text{rec}}\|_1, \quad \mathcal{L}_{\text{per}} = \|\Phi(I_h) - \Phi(I_h^{\text{rec}})\|_2^2, \quad \mathcal{L}_{\text{adv}} = [\log D(I_h) + \log(1 - D(I_h^{\text{rec}}))],$$
$$\mathcal{L}_{\text{code}} = \|\text{sg}(Z_h) - Z_h^c\|_2^2 + \beta\|Z_h - \text{sg}(Z_h^c)\|_2^2, \tag{2}$$

where the feature extractor of VGG19 (Simonyan & Zisserman, 2014) is represented by $\Phi$, $D$ is a patch-based discriminator (Isola et al., 2017), and $\text{sg}(\cdot)$ denotes the stop-gradient operator. The value of $\beta$ is set at 0.25. The final loss $\mathcal{L}_{\text{codebook}}$ is:

$$\mathcal{L}_{\text{codebook}} = \mathcal{L}_1 + \lambda_{\text{per}} \cdot \mathcal{L}_{\text{per}} + \lambda_{\text{adv}} \cdot \mathcal{L}_{\text{adv}} + \mathcal{L}_{\text{code}}, \tag{3}$$

where we set $\lambda_{\text{per}} = 1.0$ and $\lambda_{\text{adv}} = 0.8$ in our setting.

## 3.2 ASSOCIATION STAGE

In this work, we reduce the domain gap between the HQ and LQ domains, allowing the two encoders to encompass a greater range of information from both domains. Once we obtain the domain encoders from the codebook learning stage, we take inspiration from the CLIP model (Radford et al., 2021) and propose a feature patch association algorithm. This technique involves applying the feature patch association on the output from the two encoders. By utilizing this approach, we aim to further minimize the domain gap between the HQ and LQ domains.

As shown in Fig. 2(b), after obtaining the HQ and LQ domain encoder ($E_H$ and $E_L$) from the previous stage, we proceed to flatten the output features ($Z_h$ and $Z_l \in \mathbb{R}^{m \times n \times d}$) into corresponding patches ($P_i^H$ and $P_i^L, i \in [1, ..., m \times n]$). This flattening enables us to construct a similarity matrix ($M_{\text{assoc}} \in \mathbb{R}^{N \times N}, N = m \times n$), which we use to quantify the similarity between different patch features. Specifically, we calculate the cosine similarity for each patch feature and constrain them to maximize the similarity along the diagonal of the matrix. By applying this constraint, the two encoders are prompted to connect patch features close in both spatial location and feature level, preserving their spatial relationship throughout the association process. Combining the patch features from both encoders results in two associated encoders, denoted as $E_H^A$ and $E_L^A$, that integrate specific domain information, which will be utilized in the subsequent stage.

**Training Objectives.** We perform joint training on the HQ and LQ reconstruction paths, incorporating the association part. To facilitate the feature association process, we adopt the cross-entropy loss ($\mathcal{L}_{\text{CE}}^H$ and $\mathcal{L}_{\text{CE}}^L$) to effectively constrain the similarity matrix $M_{\text{assoc}}$:

$$\mathcal{L}_{\text{CE}}^H = -\frac{1}{N} \sum_{i=1}^{N} \sum_{j=1}^{C} y_{i,j} \log(p_{i,j}^h), \quad \mathcal{L}_{\text{CE}}^L = -\frac{1}{N} \sum_{i=1}^{N} \sum_{j=1}^{C} y_{i,j} \log(p_{i,j}^l), \quad (4)$$

where $N$ denotes the patch size, and $C$ represents the number of classes, which, in our specific case, is equal to $N$. The ground truth label is denoted as $y_{i,j}$, whereas the cosine similarity score in the similarity matrix $M_{\text{assoc}}$ in the HQ axis is represented as $p_{i,j}^h$. Conversely, the score in the LQ axis is defined as $p_{i,j}^l$. The final objective of the feature association part is:

$$\mathcal{L}_{\text{assoc}} = \mathcal{L}_1 + \lambda_{\text{per}} \cdot \mathcal{L}_{\text{per}} + \lambda_{\text{adv}} \cdot \mathcal{L}_{\text{adv}} + \mathcal{L}_{\text{code}} + (\mathcal{L}_{\text{CE}}^H + \mathcal{L}_{\text{CE}}^L)/2, \quad (5)$$

where we integrate the same losses and weights used in the codebook learning stage to maintain the representation of features.

## 3.3 FEATURE FUSION & CODE PREDICTION STAGE

After obtaining the two associated encoders $E_H^A$ and $E_L^A$ from the feature association stage, we use both encoders to encode the LQ image $I_l$, as shown in Fig. 2(c). Specifically, we extract feature information $Z_h^A \in \mathbb{R}^{m \times n \times d}$ and $Z_l^A \in \mathbb{R}^{m \times n \times d}$ from the LQ image $I_l$ using each encoder $E_H^A$ and $E_L^A$, separately. Similar to (Vaswani et al., 2017; Wang et al., 2022b), we use a multi-head cross-attention (MHCA) module to merge the feature information from both encoders and generate a fused feature $Z_f^A \in \mathbb{R}^{m \times n \times d}$ that incorporates the LQ domain information. This fused feature $Z_f^A$ is expected to contain useful information from both HQ and LQ domains: $Z_f^A = \text{MHCA}(Z_h^A, Z_l^A)$. The MHCA mechanism lets the module focus on different aspects of the feature space and better capture the relevant information from both encoders.

Once the fused feature $Z_f^A$ is obtained using the feature fusion technique with MHCA, we utilize a transformer-based classification approach (Zhou et al., 2022), to predict the corresponding class as code index **s**. Initially, we flatten the fused feature $Z_f^A \in \mathbb{R}^{m \times n \times d}$ as $\hat{Z}_f^A \in \mathbb{R}^{(m \cdot n) \times d}$ and input the flattened fused feature $\hat{Z}_f^A$ into the transformer and obtain the predicted code index $\mathbf{s} \in \{0, \cdots, N-1\}^{m \cdot n}$. During this process, the HQ codebook $\mathbb{C}_h$ and HQ decoder $D_H$ from the codebook learning stage are frozen. We then use the predicted code index **s** to locate the corresponding feature in the HQ codebook $\mathbb{C}_h$ and feed the resulting feature $Z_f^c$ to the decoder $D_H$ to generate HQ images $I_{res}$, as depicted in Fig. 2(c). This step efficiently enhances the image restoration by incorporating information from the hybrid domain.

**Training Objectives.** We utilize two losses to train the MHCA and transformer module effectively to ensure proper feature fusion and code index prediction learning. The first loss is an L2 loss $\mathcal{L}_{\text{code}}^{\text{feat}}$,

which encourages the fused feature $Z_f^A$ to closely resemble the quantized feature $Z_h^c$ from the HQ codebook $\mathbb{C}_h$. This loss helps ensure that the features are properly combined and maintains the relevant information from both the HQ and LQ domains. The second loss is a cross-entropy loss $\mathcal{L}_{\text{code}}^{\text{index}}$ for code index prediction, enabling the model to accurately predict the corresponding code index $\mathbf{s}$ in the HQ codebook $\mathbb{C}_h$.

$$\mathcal{L}_{\text{code}}^{\text{feat}} = \|Z_f^A - \text{sg}(Z_h^c)\|_2^2, \quad \mathcal{L}_{\text{code}}^{\text{index}} = \sum_{i=0}^{mn-1} -\hat{s}_i \log(s_i), \tag{6}$$

where we obtain the ground truth feature $Z_h^c$ and code index $\hat{\mathbf{s}}$ from the codebook learning stage, which we retrieve the quantized feature $Z_h^c$ from the HQ codebook $\mathbb{C}_h$ using the code index $\hat{\mathbf{s}}$. The final objective of the feature fusion and code prediction is:

$$\mathcal{L}_{\text{predict}} = \lambda_{\text{feat}} \cdot \mathcal{L}_{\text{code}}^{\text{feat}} + \mathcal{L}_{\text{code}}^{\text{index}}, \tag{7}$$

where we set the L2 loss weight $\lambda_{\text{feat}} = 10$ in our experiments.

## 4 EXPERIMENTS

### 4.1 EXPERIMENTAL SETUP

**Implementation Details.** In our implementation, the size of the input face image is $512 \times 512 \times 3$, and the size of the quantized feature is $16 \times 16 \times 256$. The codebooks contain $N = 1,024$ code items, and the channel of each item is 256. Throughout the entire training process, we employ the Adam optimizer (Kingma & Ba, 2014) with a batch size 32 and set the learning rate to $1.44 \times 10^{-4}$. The HQ and LQ reconstruction codebook priors are trained for 700K and 400K iterations, respectively, and the feature association part is trained for 70K iterations. Finally, the feature fusion and code prediction stage is trained for 100K iterations. The proposed method is implemented in Pytorch and trained with eight NVIDIA Tesla A100 GPUs.

**Training Dataset.** We train our model on the FFHQ dataset (Karras et al., 2019), which contains 70,000 high-quality face images. For training, we resize all images from $1024 \times 1024$ to $512 \times 512$. To generate the paired data, we synthesize the degraded images on the FFHQ dataset using the same procedure as the compared methods (Li et al., 2018; 2020; Wang et al., 2021; 2022b; Gu et al., 2022; Zhou et al., 2022). First, the HQ image $I_{\text{high}}$ is blurred (convolution operator $\otimes$) by a Gaussian kernel $k_\sigma$. Subsequently, a downsampling operation $\downarrow$ with a scaling factor $r$ is performed to reduce the image's resolution. Next, additive Gaussian noise $n_\delta$ is added to the downsampled image. JPEG compression with a quality factor $q$ is applied to further degrade image quality. The resulting image is then upsampled $\uparrow$ with a scaling factor $r$ to a resolution of $512 \times 512$ to obtain the degraded $I_{\text{low}}$ image. In our experiment setting, we randomly sample $\sigma$, $r$, $\delta$, and $q$ from [0.1, 15], [0.8, 30], [0, 20], and [30, 100], respectively. The procedure can be formulated as follows:

$$I_{\text{low}} = \{[(I_{\text{high}} \otimes k_\sigma) \downarrow_r + n_\delta] \text{JPEG}_q\} \uparrow_r. \tag{8}$$

**Testing Dataset.** Our evaluation follows the settings in prior literature (Wang et al., 2022b; Gu et al., 2022; Zhou et al., 2022), and includes four datasets: the synthetic dataset CelebA-Test and three real-world datasets, namely, LFW-Test, WIDER-Test, and BRIAR-Test. CelebA-Test comprises 3,000 images selected from the CelebA-HQ testing partition (Karras et al., 2018). LFW-Test consists of 1,711 images representing the first image of each identity in the validation part of the LFW dataset (Huang et al., 2008). Zhou *et al.* (Zhou et al., 2022) collected the WIDER-Test from the WIDER Face dataset (Yang et al., 2016), which comprises 970 face images. Lastly, the BRIAR-Test contains 2,120 face images selected from the BRIAR dataset (Cornett et al., 2023). The BRIAR-Test dataset provides a wider range of challenging and diverse degradation levels that enable the evaluation of the generalization and robustness of face restoration methods (All subjects shown in the paper have consented to publication.).

**Metrics.** In evaluating our method's performance on the CelebA-Test dataset with ground truth, we employ PSNR, SSIM, and LPIPS (Zhang et al., 2018) as evaluation metrics. We utilize the commonly used non-reference perceptual metrics, FID (Heusel et al., 2017) and NIQE (Mittal et al., 2012) to evaluate real-world datasets without ground truth. Similar to prior work (Gu et al., 2022; Wang et al., 2022b; Zhou et al., 2022), we measure the identity of the generated images by using the embedding angle of ArcFace (Deng et al., 2019), referred to as "IDA". To better measure the fidelity of generated facial images with accurate facial positions and expressions, we additionally adopt landmark distance (LMD) as the fidelity metric.

Table 1: **Quantitative comparisons.** Red and blue indicate the best and second-best, respectively.

(a) *Real-world* datasets.

| Dataset | LFW-Test | | WIDER-Test | | BRIAR-Test | |
|---|---|---|---|---|---|---|
| Methods | FID↓ | NIQE↓ | FID↓ | NIQE↓ | FID↓ | NIQE↓ |
| Input | 137.587 | 11.003 | 199.972 | 13.498 | 201.061 | 10.784 |
| PSFRGAN | 49.551 | 4.094 | 49.857 | 4.033 | 196.774 | 3.979 |
| GFP-GAN | 50.057 | 3.966 | 39.730 | 3.885 | 97.360 | 5.281 |
| GPEN | 51.942 | 3.902 | 46.359 | 4.104 | 91.653 | 5.166 |
| RestoreFormer | 48.412 | 4.168 | 49.839 | 3.894 | 107.654 | 5.064 |
| CodeFormer | 52.350 | 4.482 | 38.798 | 4.164 | 98.134 | 5.018 |
| VQFR | 50.712 | 3.589 | 44.158 | 3.054 | 92.072 | 4.970 |
| DR2 | 46.550 | 5.150 | 45.726 | 5.188 | 96.968 | 5.417 |
| **DAEFR (Ours)** | 47.532 | 3.552 | 36.720 | 3.655 | 90.032 | 4.649 |

(b) The *synthetic* CelebA-Test dataset.

| Methods | FID↓ | LPIPS↓ | NIQE↓ | IDA↓ | LMD↓ | PSNR↑ | SSIM↑ |
|---|---|---|---|---|---|---|---|
| Input | 337.013 | 0.528 | 19.287 | 1.426 | 17.016 | 20.833 | 0.638 |
| PSFRGAN | 66.367 | 0.450 | 3.811 | 1.260 | 7.713 | 20.303 | 0.536 |
| GFP-GAN | 46.130 | 0.453 | 4.061 | 1.268 | 9.501 | 19.574 | 0.522 |
| GPEN | 55.308 | 0.425 | 3.913 | 1.141 | 7.259 | 20.545 | 0.552 |
| RestoreFormer | 54.395 | 0.467 | 4.013 | 1.231 | 8.883 | 20.146 | 0.494 |
| CodeFormer | 62.021 | 0.365 | 4.570 | 1.049 | 5.381 | 21.449 | 0.575 |
| VQFR | 54.010 | 0.456 | 3.328 | 1.237 | 9.128 | 19.484 | 0.472 |
| DR2 | 63.675 | 0.409 | 5.104 | 1.215 | 7.890 | 20.327 | 0.595 |
| **DAEFR (Ours)** | 52.056 | 0.388 | 4.477 | 1.071 | 5.634 | 19.919 | 0.553 |

Figure 3: **Qualitative comparison on *real-world* datasets.** The BRIAR-Test dataset contains original identity clean images, allowing us to ascertain that the individual in this image is not wearing glasses. Our DAEFR method exhibits robustness in restoring high-quality faces even under heavy degradation.

## 4.2 COMPARISONS WITH STATE-OF-THE-ART METHODS

We compare the proposed method, DAEFR, with state-of-the-art methods: PSFRGAN (Chen et al., 2021), GFP-GAN (Wang et al., 2021), GPEN (Yang et al., 2021), RestoreFormer (Wang et al., 2022b), CodeFormer (Zhou et al., 2022), VQFR (Gu et al., 2022), and DR2 (Wang et al., 2023).

**Evaluation on Real-world Datasets.** As shown in Table 1(a), our proposed DAEFR outperforms other methods regarding perceptual quality metrics, evidenced by the lowest FID scores on the WIDER-Test and BRIAR-Test dataset and second-best scores on the LFW-Test dataset, indicating the statistical similarity between the distributions of real and generated images. Regarding NIQE scores, which assess the perceptual quality of images, our method achieves the highest score on the LFW-Test dataset and the second-highest on the WIDER-Test and BRIAR-Test datasets.

Visual comparisons in Fig. 3 further demonstrate the robustness of our method against severe degradation, resulting in visually appealing outcomes. In contrast, RestoreFormer (Wang et al., 2022b) and VQFR (Gu et al., 2022) exhibit noticeable artifacts in their results, while CodeFormer (Zhou et al., 2022) and DR2 (Wang et al., 2023) tend to produce smoothed outcomes, losing intricate facial details. DAEFR, however, effectively preserves the identity of the restored images, producing natural results with fine details. This preservation of identity can be attributed to the design of our model, which emphasizes the retention of original facial features during the restoration process. These observations underscore the strong generalization ability of our method.

**Evaluation on the Synthetic Dataset.** We compare DAEFR quantitatively with existing approaches on the synthetic CelebA-Test dataset in Table 1(b). Our method demonstrates competitive performance in image quality metrics, namely FID and LPIPS, achieving the second-best scores among the evaluated methods. These metrics represent the distribution and visual similarity between the restored and original images. Moreover, our approach effectively preserves identity information, as evidenced by comparable IDA and LMD scores. These metrics assess how well the model preserves the identity and facial structure of the original image.

To further illustrate these points, we provide a qualitative comparison in Fig. 4. Our method demonstrates the ability to generate high-quality restoration results with fine facial details, surpassing the performance of the other methods. This ability to restore fine facial details results from our model's capacity to extract and utilize high-frequency information from the degraded images, contributing to its overall performance. We provide more visual comparisons in the appendix and supplementary.

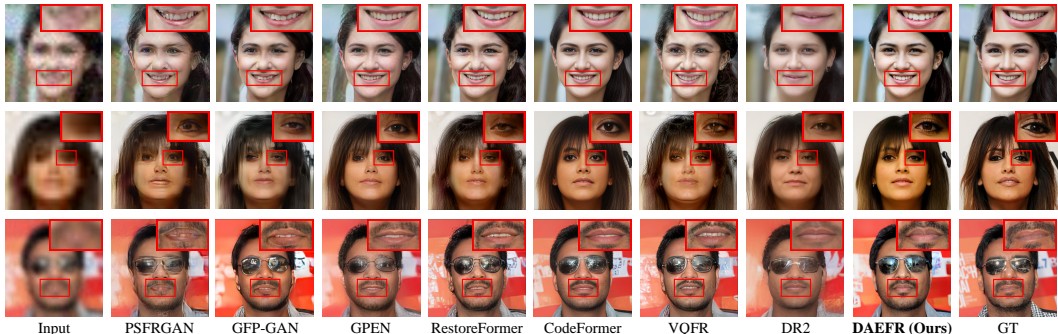

Input  PSFRGAN  GFP-GAN  GPEN  RestoreFormer  CodeFormer  VQFR  DR2  **DAEFR (Ours)**  GT

Figure 4: **Qualitative comparison on the *synthetic* CelebA-Test dataset.** Our DAEFR method exhibits robustness in restoring high-quality faces even under heavy degradation.

Table 2: **Ablation studies of variant networks and association methods on the CelebA-Test.** The terms "HQ encode" and "LQ encoder" refer to the encoders used for encoding LQ images. "w/o" and "with" in the association part indicate whether the encoder undergoes the feature association process. The fusion types "Linear" and "MHCA" in the feature fusion module represent linear projection with three fully connected layers or multi-head cross attention, respectively.

| Exp. | Encoder (Sec. 3.1) | | Association (Sec. 3.2) | | Feature Fusion (Sec. 3.3) | | Metrics | | |
|---|---|---|---|---|---|---|---|---|---|
| | HQ encoder | LQ encoder | w/o | with | Linear | MHCA | LPIPS↓ | LMD↓ | NIQE↓ |
| (a) | ✓ | | | | | | 0.344 | 4.170 | 4.394 |
| (b) | | ✓ | | | | | 0.343 | 4.265 | 4.510 |
| (c) | ✓ | ✓ | ✓ | | ✓ | | 0.349 | 4.258 | 4.297 |
| (d) | ✓ | ✓ | | ✓ | ✓ | | **0.343** | 4.197 | 4.290 |
| (e) (Ours) | ✓ | ✓ | | ✓ | | ✓ | 0.351 | **4.019** | **3.815** |

## 4.3 ABLATION STUDY

**Number of Encoders.**  In our research, we conduct an initial investigation to examine the impact of the number of encoders on the overall performance. We present this analysis in Exp. (a) to (c) as detailed in Table 2. The results of our investigation indicate that utilizing two encoders as the input source yields better performance than using a single encoder. This conclusion is supported by the superior NIQE score achieved in the experiments. This ablation study confirms that including an additional LQ encoder can provide domain-specific information, aiding image restoration. To provide further insights, we show the visual results of Exp. (a) to (c) in Fig. 5.

**Association Stage.**  Furthermore, we evaluate the effectiveness of our association stage through Exp. (c) and (d), as depicted in Table 2. The results indicate that utilizing associated encoders as our input source leads to superior performance across all evaluation metrics compared to non-associated encoders. This empirical evidence validates that our association stage effectively enhances the encoder's capability to retrieve information from the alternative domain and effectively reduces the domain gap between HQ and LQ. To provide visual evidence of the improvements achieved by our approach, we present the visual results of Exp. (c) and (d) in Fig. 5. These visual comparisons further support the superiority of our method.

**Feature Fusion Module.**  Finally, we assess the effectiveness of our feature fusion module, comparing two different approaches: a baseline method employing linear projection with three fully connected layers and our proposed multi-head cross-attention module (MHCA). The experimental results are presented in Table 2, specifically under Exp. (d) and Exp. (e). We aim to determine which approach yields better performance regarding both LMD and NIQE scores. The results demonstrate a notable improvement when utilizing the MHCA compared to the straightforward linear projection method for feature fusion. This indicates that our MHCA can effectively fuse domain-specific information from both the HQ and LQ domains while preserving crucial identity information. To provide a visual representation of the results, we showcase the outcomes of Exp. (d) and Exp. (e) in Fig. 5. These visual representations further support our findings and demonstrate the superior performance of our proposed MHCA in enhancing the image restoration process.

**Effectiveness of Low-Quality Feature from Auxiliary Branch.**  To demonstrate the effectiveness of our auxiliary LQ branch, we conduct validated experiments of fusing LQ features with feature $Z_f^c$ in the feature fusion and code prediction stage. These experiments involve extracting LQ features from the LQ codebook and adding a control module (Wang et al., 2018; Zhou et al., 2022) to fuse the LQ feature and feature $Z_f^c$ before feeding to the HQ decoder.

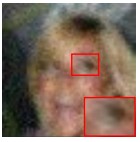 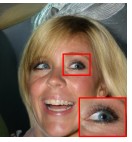 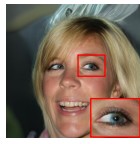 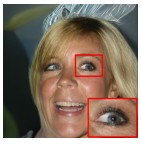 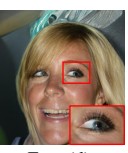 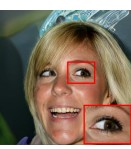 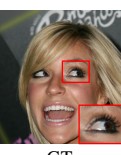

| Input | Exp. (a) | Exp. (b) | Exp. (c) | Exp. (d) | Exp. (e) Ours | GT |

Figure 5: **Ablation studies.** The experimental index in accordance with the Table 2 configuration is utilized. Our method successfully produces intricate facial details and closely resembles the ground truth, even when the input undergoes severe degradation. Importantly, we effectively retain the identity information from the degraded input.

Table 3: **Quantitative evaluation for the effectiveness of LQ feature on the CelebA-Test.** The LQ feature scale **s** indicates the fusion scalar with feature $Z_f^c$. The performance becomes better when we increase the scalar $s_{lq}$, and the experimental results prove the effectiveness of our LQ feature, which encodes essential visual attributes and domain-specific statistical characteristics.

| LQ Feature Scale | $s_{lq}$=0 | $s_{lq}$=0.2 | $s_{lq}$=0.4 | $s_{lq}$=0.6 | $s_{lq}$=0.8 | $s_{lq}$=1.0 |
|---|---|---|---|---|---|---|
| LPIPS ↓ | 0.399 | 0.389 | 0.378 | 0.370 | 0.364 | **0.363** |
| PSNR ↑ | 20.040 | 20.367 | 20.727 | 21.074 | 21.348 | **21.488** |
| SSIM ↑ | 0.558 | 0.571 | 0.585 | 0.600 | 0.612 | **0.616** |

Our quantitative evaluation employs quality metrics, including LPIPS, PSNR, and SSIM. We conduct these experiments on the CelebA-Test dataset. Our quantitative results clearly prove the positive impact when we increase the scale of LQ feature scalar $s_{lq}$, as shown in Table 3. This experiment underscores the practical advantages of our LQ branch and shows the LQ features effectively encode essential visual attributes and domain-specific characteristics, enhancing the image restoration process. We place the detailed experiment setting and visual comparison in the appendix and supplementary.

Table 4: **Quantitative evaluation on the face recognition task.** We conduct the quantitative experiments on the LFW dataset (Huang et al., 2008) of the face recognition task with the official ArcFace (Deng et al., 2019) model with Verification performance (%) as our evaluation metric. The degradation parameters ranging from 10,000 to 40,000 correspond to varying levels of degradation.

| Methods | Origin | 10000 (Slight) | 20000 | 30000 | 40000 (Severe) |
|---|---|---|---|---|---|
| CodeFormer (Zhou et al., 2022) | 98.23 | 97.35 | 91.28 | 81.25 | 71.67 |
| **DAEFR (Ours)** | **98.51** | **97.86** | **92.15** | **82.69** | **73.78** |

**Validation on Downstream Face Recognition Task.** We conduct a downstream face recognition task using the proposed restoration method on the LFW (Huang et al., 2008) face recognition validation split set (with 12,000 images). We use the unseen atmospheric turbulence degradation to simulate diverse degradation levels, employing the methodology outlined in (Chimitt & Chan, 2020) for degradation generation. The degradation parameters ranging from 10,000 to 40,000 correspond to varying levels of degradation, spanning from slight to severe. We provide the dataset samples in the appendix and supplementary material.

We employ the official ArcFace (Deng et al., 2019) model with Verification performance (%) as our evaluation metric to evaluate the restored images. The experimental results, presented in Table 4, demonstrate the superior performance of our method across various degradation levels. Particularly noteworthy is the widening performance gap between CodeFormer and our method as the degradation severity escalates. These findings validate the efficacy of our additional LQ encoder, which captures valuable information from the LQ domain, thus significantly augmenting the restoration process.

## 5 CONCLUSION

In this paper, we propose DAEFR to effectively tackle the challenge of blind face restoration, generating high-quality facial images from low-quality ones despite the domain gap and information loss between HQ and LQ image types. We introduce an auxiliary LQ encoder that captures the LQ domain's specific visual characteristics and statistical properties. We also employ feature association techniques between the HQ and LQ encoders to alleviate the domain gap. Furthermore, we leverage attention mechanisms to fuse the features extracted from the two associated encoders, enabling them to contribute to the code prediction process and produce high-quality results. Our experiments show that DAEFR produces promising results in both synthetic and real-world datasets with severe degradation, demonstrating its effectiveness in blind face restoration.

## 6 ACKNOWLEDGEMENT

This research is based upon work supported by the Office of the Director of National Intelligence (ODNI), Intelligence Advanced Research Projects Activity (IARPA), via IARPA R&D Contract No. 2022-21102100001. The views and conclusions contained herein are those of the authors and should not be interpreted as necessarily representing the official policies or endorsements, either expressed or implied, of the ODNI, IARPA, or the U.S. Government. The U.S. Government is authorized to reproduce and distribute reprints for Governmental purposes notwithstanding any copyright annotation thereon.

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

# Appendix

## A OVERVIEW

This supplementary material presents additional results to complement the main manuscript. First, we describe the detailed network architecture in Section B. Then, we conduct more ablation studies to demonstrate the effectiveness of the association stage and our auxiliary LQ branch in Section C. Finally, we show more visual comparisons with state-of-the-art methods in Section D.

## B NETWORK ARCHITECTURE

The detailed structures of the encoder and decoder are shown in Table 5. Our restoration process employs an identical encoder and decoder structure for both the HQ and LQ paths. We use a transformer (Vaswani et al., 2017) module consisting of nine self-attention blocks for the structure. Additionally, we enhance its expressiveness by incorporating sinusoidal positional embedding (Carion et al., 2020; Cheng et al., 2022) on query $\mathbf{Q}$ and keys $\mathbf{K}$. This modification aids in capturing positional information effectively within the transformer. In the feature fusion stage, we employ a multi-head cross-attention module (MHCA) to effectively merge the HQ and LQ features extracted from the encoders, which is inspired by (Wang et al., 2022b; Vaswani et al., 2017). The implementation of MHCA is:

$$\mathbf{Q} = Z_l^A \mathbf{W}_q + \mathbf{b}_q, \quad \mathbf{K} = Z_h^A \mathbf{W}_k + \mathbf{b}_k, \quad \mathbf{V} = Z_h^A \mathbf{W}_v + \mathbf{b}_v, \qquad (9)$$

where we utilize feature $Z_l^A$ as queries $\mathbf{Q}$ and feature $Z_h^A$ as keys $\mathbf{K}$ and values $\mathbf{V}$. These features are multiplied by their respective learnable weights $\mathbf{W}_{q/k/v} \in \mathbb{R}^{d \times d}$ and biased by $\mathbf{b}_{q/k/v} \in \mathbb{R}^d$.

$$Z_f^A = \text{MHCA}(Z_h^A, Z_l^A) = \text{FFN}(\text{LN}(Z_h^A, Z_l^A)), \qquad (10)$$

where LN is the layer normalization, and FFN is the feed-forward network composed of two convolution layers.

Table 5: **Detailed architecture of our encoder and decoder.** GN: GroupNorm; c: channels; $\mathbf{C}$ is the length of the features in the HQ codebook. ↓ and ↑ mean the feature pass direction in the table.

| Input size | Encoder ↓ | Decoder ↑ |
|---|---|---|
| 512×512 | { {Residual block: 32-GN, 64-c} × 2 } × 2
Bilinear downsampling 2×
Conv 3 × 3, 64-c | { {Residual block: 32-GN, 64-c} × 2 } × 3
Conv 3 × 3, 128-c
Bilinear upsampling 2× |
| 256×256 | { {Residual block: 32-GN, 128-c} × 2 } × 2
Bilinear downsampling 2×
Conv 3 × 3, 128-c | { {Residual block: 32-GN, 128-c} × 2 } × 3
Conv 3 × 3, 128-c
Bilinear upsampling 2× |
| 128×128 | { {Residual block: 32-GN, 128-c} × 2 } × 2
Bilinear downsampling 2×
Conv 3 × 3, 128-c | { {Residual block: 32-GN, 128-c} × 2 } × 3
Conv 3 × 3, 256-c
Bilinear upsampling 2× |
| 64×64 | { {Residual block: 32-GN, 256-c} × 2 } × 2
Bilinear downsampling 2×
Conv 3 × 3, 256-c | { {Residual block: 32-GN, 256-c} × 2 } × 3
Conv 3 × 3, 256-c
Bilinear upsampling 2× |
| 32×32 | { {Residual block: 32-GN, 256-c} × 2 } × 2
Bilinear downsampling 2×
Conv 3 × 3, 256-c | { {Residual block: 32-GN, 256-c} × 2 } × 3
Conv 3 × 3, 512-c
Bilinear upsampling 2× |
| 16×16 | { {Residual block: 32-GN, 512-c} × 2 } × 2
Conv_out
Conv 3 × 3, 512-c → $\mathbf{C}$-c | { {Residual block: 32-GN, 512-c} × 2 } × 3
Conv 3 × 3, $\mathbf{C}$-c → 512-c |

## C MORE ABLATION STUDIES

In this section, we delve into further ablation studies concerning the association stage and our auxiliary LQ branch, focusing on architecture design and various loss designs. In our proposed approach,

we aim to demonstrate the association stage's effectiveness and our auxiliary LQ branch. Through these studies, we provide detailed analysis and evidence to support the impact and contribution of the association stage and our auxiliary LQ branch in improving the overall performance of the image restoration process.

**Domain Gap before Association Stage**   We thoroughly investigate the domain gap in three distinct pathways: HQ path, LQ path, and Hybrid path. All three pathways involve the utilization of LQ images as input. Specifically, the LQ image is inputted into the LQ encoder in the Hybrid path, and the resulting features are subsequently forwarded to the HQ codebook. The HQ codebook is employed to identify the closest feature, which is then utilized for reconstruction using the HQ decoder. As illustrated in Figure 6, both the HQ path and Hybrid path exhibit limitations in effectively reconstructing or restoring the LQ image due to the domain gap. Notably, the Hybrid path fails to generate any facial features.

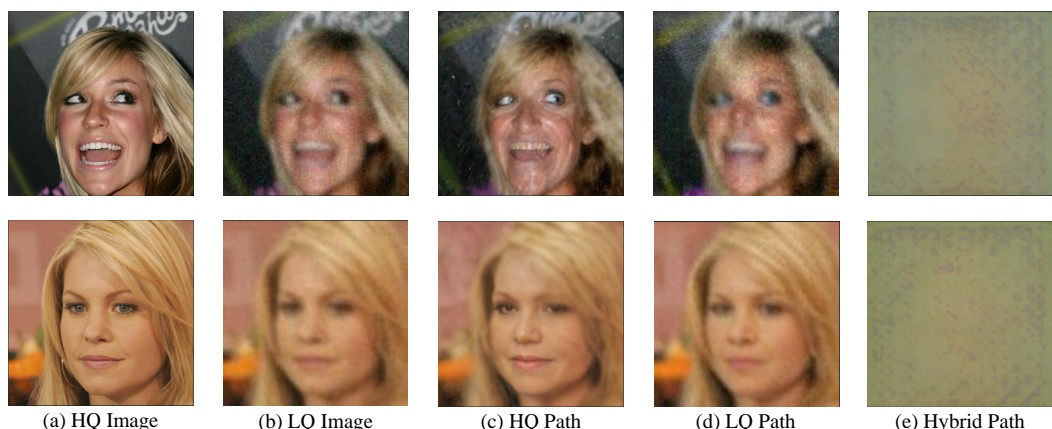

| (a) HQ Image | (b) LQ Image | (c) HQ Path | (d) LQ Path | (e) Hybrid Path |

Figure 6: **Domain gap.** We utilize **LQ images** as input to evaluate the reconstruction ability of various path types. The significant domain gap between HQ and LQ images significantly affects the ability to reconstruct the images. (Hybrid Path: LQ encoder + HQ codebook + HQ decoder)

**Different Association Architecture**   We explore various network architectures to enhance the capabilities of the LQ encoder, as depicted in Fig.7. However, our experimental results demonstrated that solely enabling the LQ encoder to learn from the HQ encoder did not effectively address the domain gap issue. Furthermore, the less constrained HQ feature representation did not align well with LQ features, leading to the generation of non-face images, as depicted in Fig.8(c). Therefore, we opt for a joint optimization approach, incorporating both the HQ and LQ paths, to preserve the feature representation. This approach aims to empower both encoders to capture more comprehensive information from both domains and reduce the domain gap.

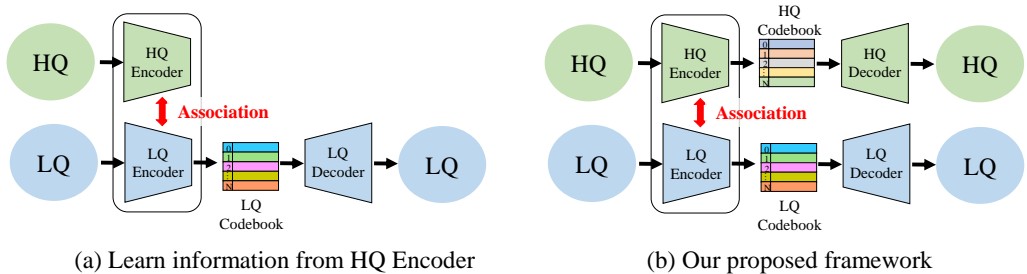

| (a) Learn information from HQ Encoder | (b) Our proposed framework |

Figure 7: **Different association architectures.** (a) Force the LQ encoder to learn information from the HQ encoder. (b) Joint optimization for both the HQ and LQ paths.

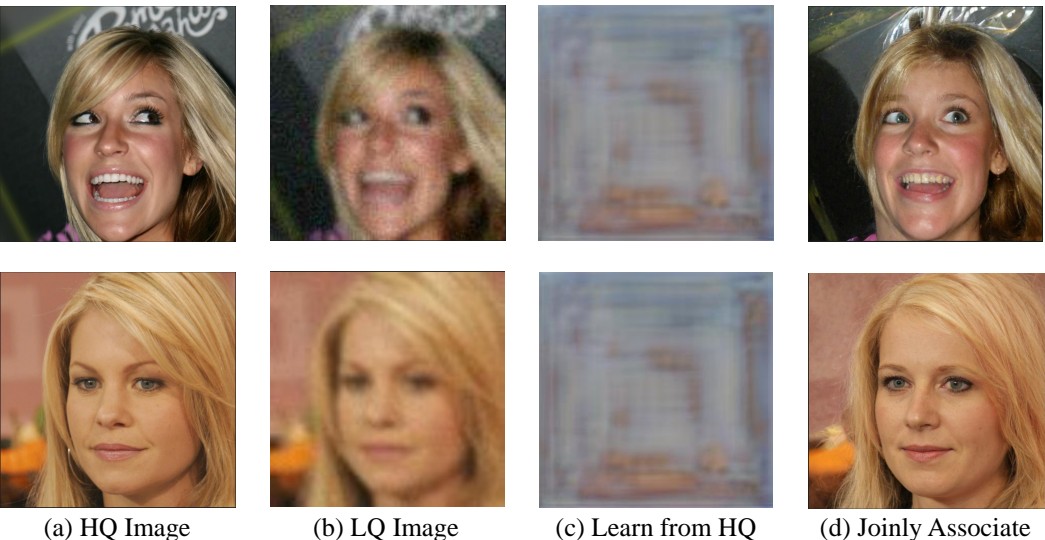

|              |              |                |                   |
| :----------: | :----------: | :------------: | :---------------: |
| (a) HQ Image | (b) LQ Image | (c) Learn from HQ | (d) Joinly Associate |

Figure 8: **Visual results for different association architectures.** We utilize LQ images as input. (c) is the result of only letting the LQ encoder learn the information from the HQ encoder. (d) is the result of jointly optimizing both paths.

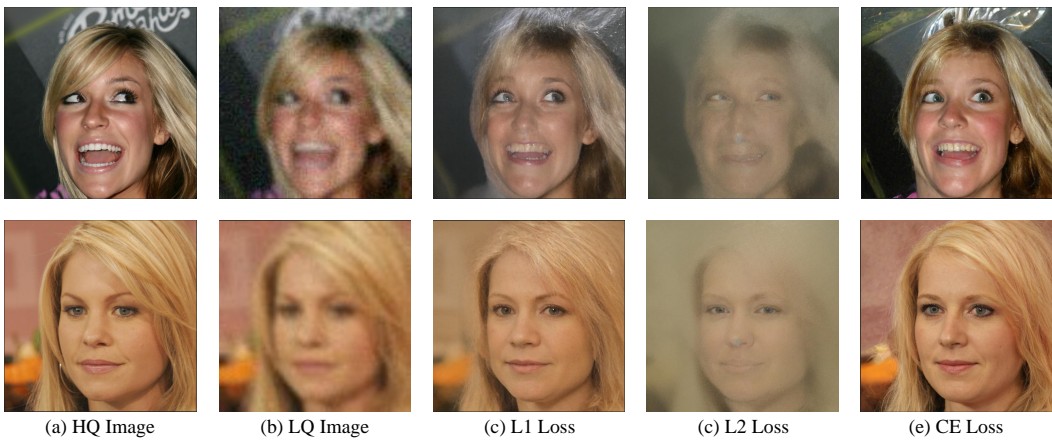

|              |              |             |             |            |
| :----------: | :----------: | :---------: | :---------: | :--------: |
| (a) HQ Image | (b) LQ Image | (c) L1 Loss | (c) L2 Loss | (e) CE Loss |

Figure 9: **Visual results for different loss settings.** We utilize LQ images as input. (c) (d) The L1 and L2 loss fails to associate the features of the two domains effectively and loses certain original information. (e) The utilization of cross-entropy loss enhances the association between the features of the two domains, resulting in the preservation of a greater amount of original information.

**Different Loss Settings**  We explore various types of loss functions to address the similarity between HQ and LQ features. We investigate using the L1 and L2 loss to enforce an exact match between the HQ and LQ features. However, the experimental results demonstrate that this approach does not yield satisfactory outcomes, as shown in Fig. 9. We also present the quantitative results of our CelebA validation dataset in Table 6.

Drawing inspiration from the CLIP model (Radford et al., 2021), we construct a similarity matrix and utilize the cross-entropy loss to constrain the feature similarity. As depicted in Fig. 9, we configure our experiments to use the LQ image as input and employ the LQ encoder to generate features. These features are then processed through the HQ codebook to identify the nearest matching feature and then decoded using the HQ decoder.

By incorporating the cross-entropy loss, we effectively establish associations between the HQ and LQ features, enhancing the overall performance of our approach.

Table 6: **Quantitative evaluation.** We conduct the quantitative experiments with different association loss setting on our CelebA validation set.

| Losses | FID↓ | LPIPS↓ |
|---|---|---|
| Input | 143.991 | 0.444 |
| L1 Loss | 70.717 | 0.403 |
| L2 Loss | 138.212 | 0.552 |
| Cross-Entropy Loss | **41.230** | **0.361** |

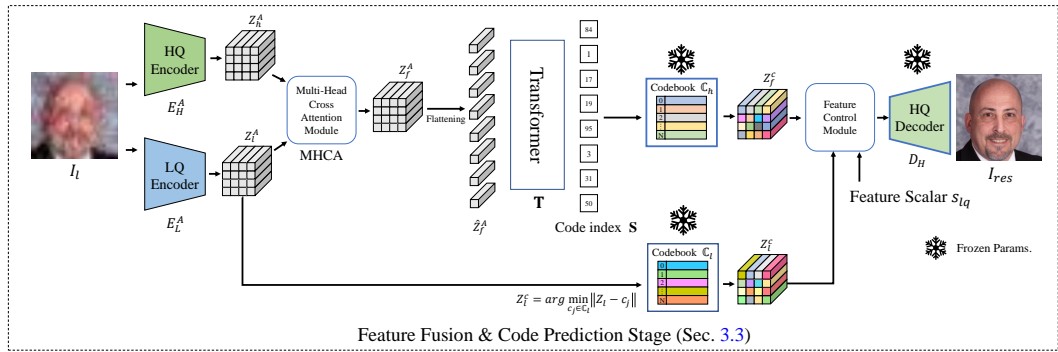

Figure 10: **Detailed network architecture for the experiment of controlling the scale of LQ features.** We conduct validated experiments of fusing LQ features in the feature fusion and code prediction stage. Given a feature control module and feature scalar $s_{lq}$, we can control the scale of the LQ feature $Z_l^c$ to fuse with the feature $Z_f^c$ before feeding to the HQ decoder.

**Effectiveness of Low-Quality Feature from Auxiliary Branch.** To demonstrate the effectiveness of our auxiliary LQ branch, we conduct validated experiments of fusing LQ features with feature $Z_f^c$ in the feature fusion and code prediction stage. These experiments involve extracting LQ features $Z_l^c$ from the LQ codebook and adding a control module (Wang et al., 2018; Zhou et al., 2022). Given a feature control module and feature scalar $s_{lq}$, we can control the scale of the LQ feature $Z_l^c$ to fuse with the feature $Z_f^c$ before feeding to the HQ decoder. The network architecture is shown in Fig. 10.

We conduct these experiments on the CelebA-Test dataset. Our visual results clearly prove the positive impact when we increase the scale of LQ feature scalar $s_{lq}$, as shown in Fig. 11. This experiment underscores the practical advantages of our LQ branch and shows the LQ features effectively encode essential visual attributes and domain-specific characteristics, enhancing the image restoration process.

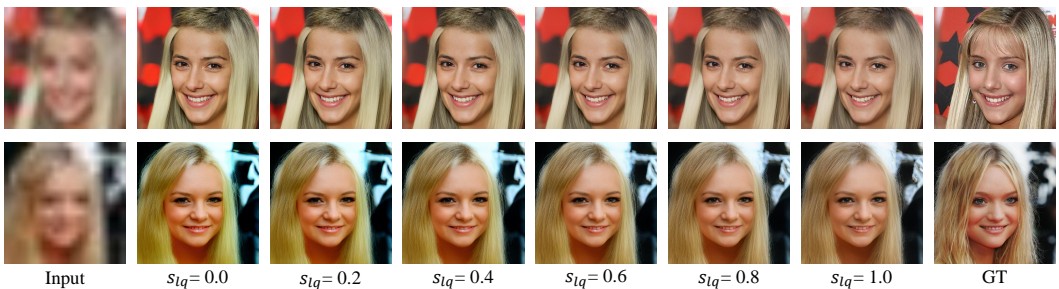

Figure 11: **Visual comparison with different LQ feature scalar.** We compare the visual results with different LQ feature scalar $s_{lq}$, showing that the LQ features indeed encode essential visual attributes and domain-specific characteristics, even when the input is severely degraded.

**Validation on Downstream Face Recognition Task** We conduct a comprehensive downstream face recognition task with the following procedure. We utilize the LFW (Huang et al., 2008) Face

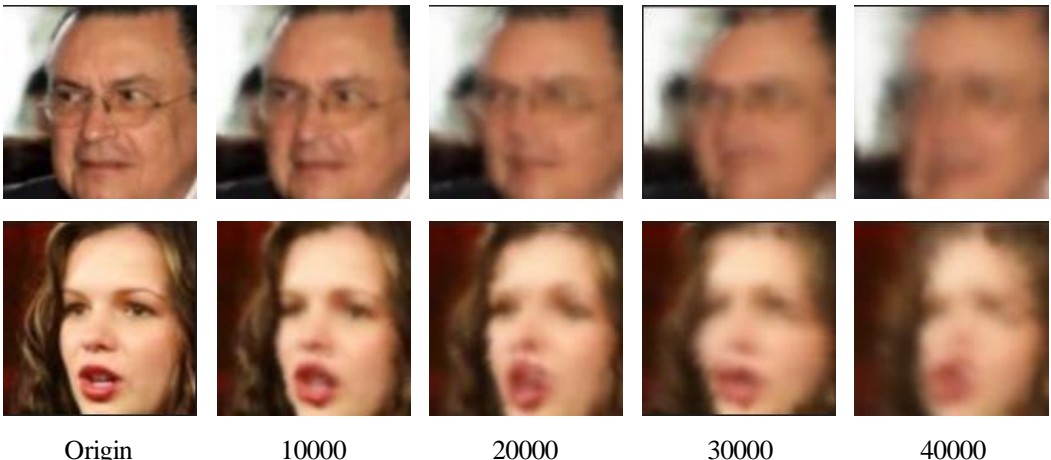

|  Origin | 10000 | 20000 | 30000 | 40000 |

Figure 12: **Visualization for face recognition validation dataset.** We choose the unseen atmospheric turbulence degradation to simulate diverse degradation levels, employing the methodology outlined in (Chimitt & Chan, 2020) for degradation generation. The degradation parameters ranging from 10,000 to 40,000 correspond to varying levels of degradation, spanning from slight to severe.

recognition dataset's validation split, comprising 12,000 images. We choose the unseen atmospheric turbulence degradation to simulate diverse degradation levels, employing the methodology outlined in (Chimitt & Chan, 2020) for degradation generation. The degradation parameters ranging from 10,000 to 40,000 correspond to varying levels of degradation, spanning from slight to severe. We provide the dataset samples in Fig. 12.

## D   MORE VISUAL COMPARISON

In this section, we present additional visual comparisons with state-of-the-art methods: PSFR-GAN (Chen et al., 2021), GFP-GAN (Wang et al., 2021), GPEN (Yang et al., 2021), Restore-Former (Wang et al., 2022b), CodeFormer (Zhou et al., 2022), VQFR (Gu et al., 2022), and DR2 (Wang et al., 2023).

The qualitative comparisons on the *LFW-Test* are shown in Fig. 13. The qualitative comparisons on the *WIDER-Test* are shown in Fig. 14, Fig. 15 and Fig. 16. The qualitative comparisons on the *BRIAR-Test* are shown in Fig. 17. The qualitative comparisons on the *CelebA-Test* are shown in Fig. 18.

We also compare the large occlusion and pose situation with state-of-the-art methods in Fig. 19.

While our method demonstrates robustness in most severe degradation scenarios, we also observe instances where it may fail, particularly in cases with large face poses. This can be expected as the FFHQ dataset contains few samples with large face poses, leading to a scarcity of relevant codebook features to effectively address such situations, resulting in less satisfactory restoration and reconstruction outcomes. We show the failure cases in Fig. 20.

These comparisons demonstrate that our proposed DAEFR generates high-quality facial images and effectively preserves the identities, even under severe degradation of the input faces. Furthermore, compared to the alternative methods, DAEFR performs better in recovering finer details and producing more realistic facial outputs.

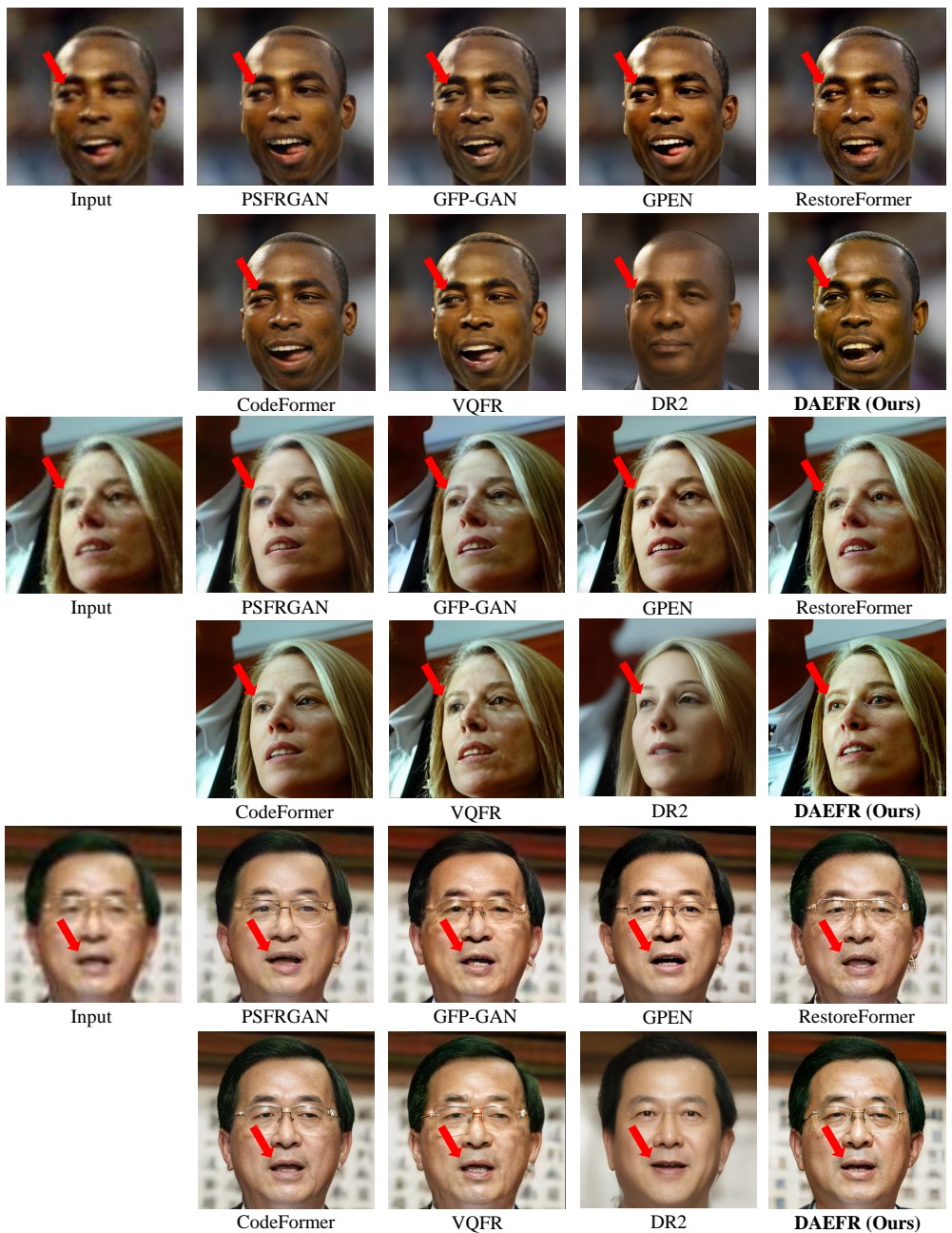

Figure 13: **Qualitative comparison on *LFW-Test* datasets.** Our DAEFR method exhibits robustness in restoring high-quality faces in detail part.

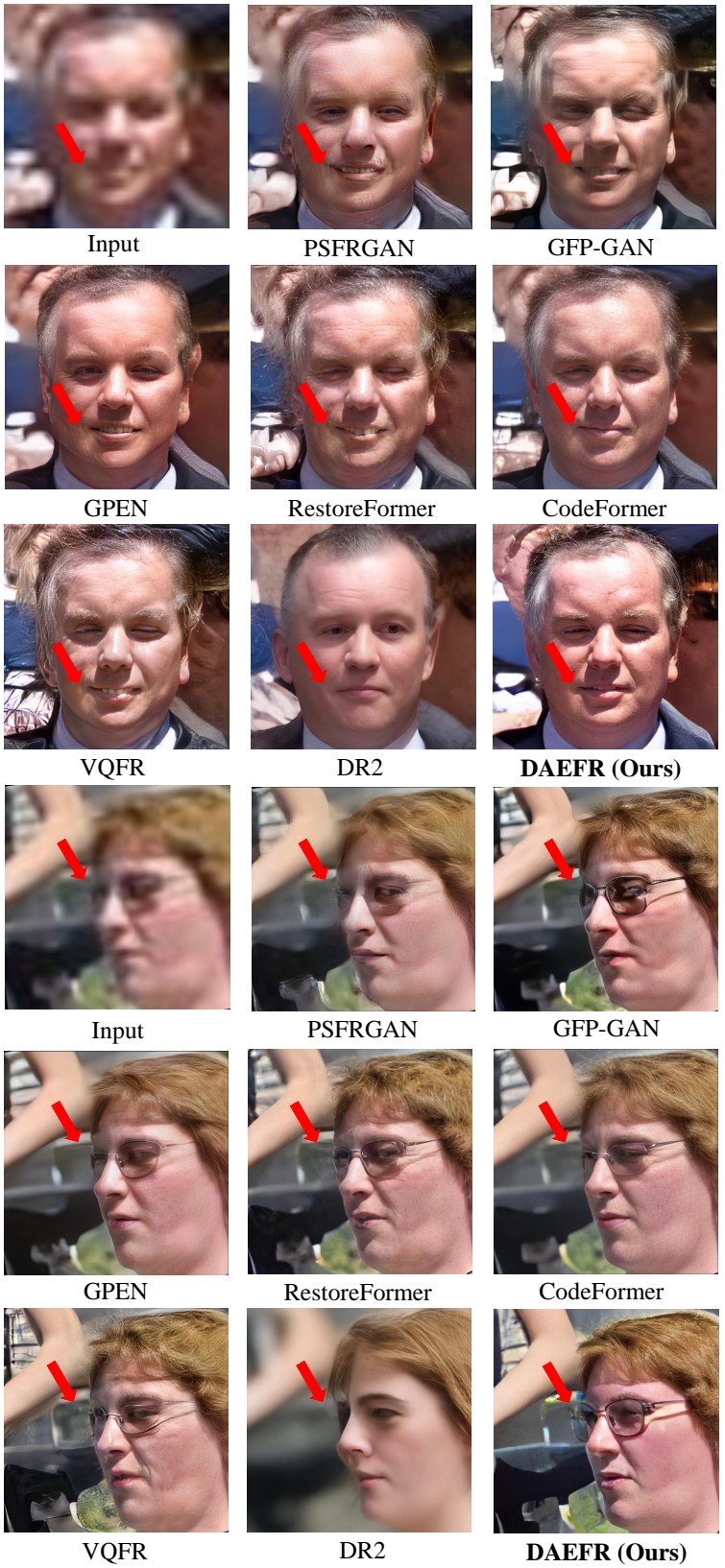

Figure 14: **Qualitative comparison on *WIDER-Test* datasets.** Our DAEFR method exhibits robustness in restoring high-quality faces in detail part.

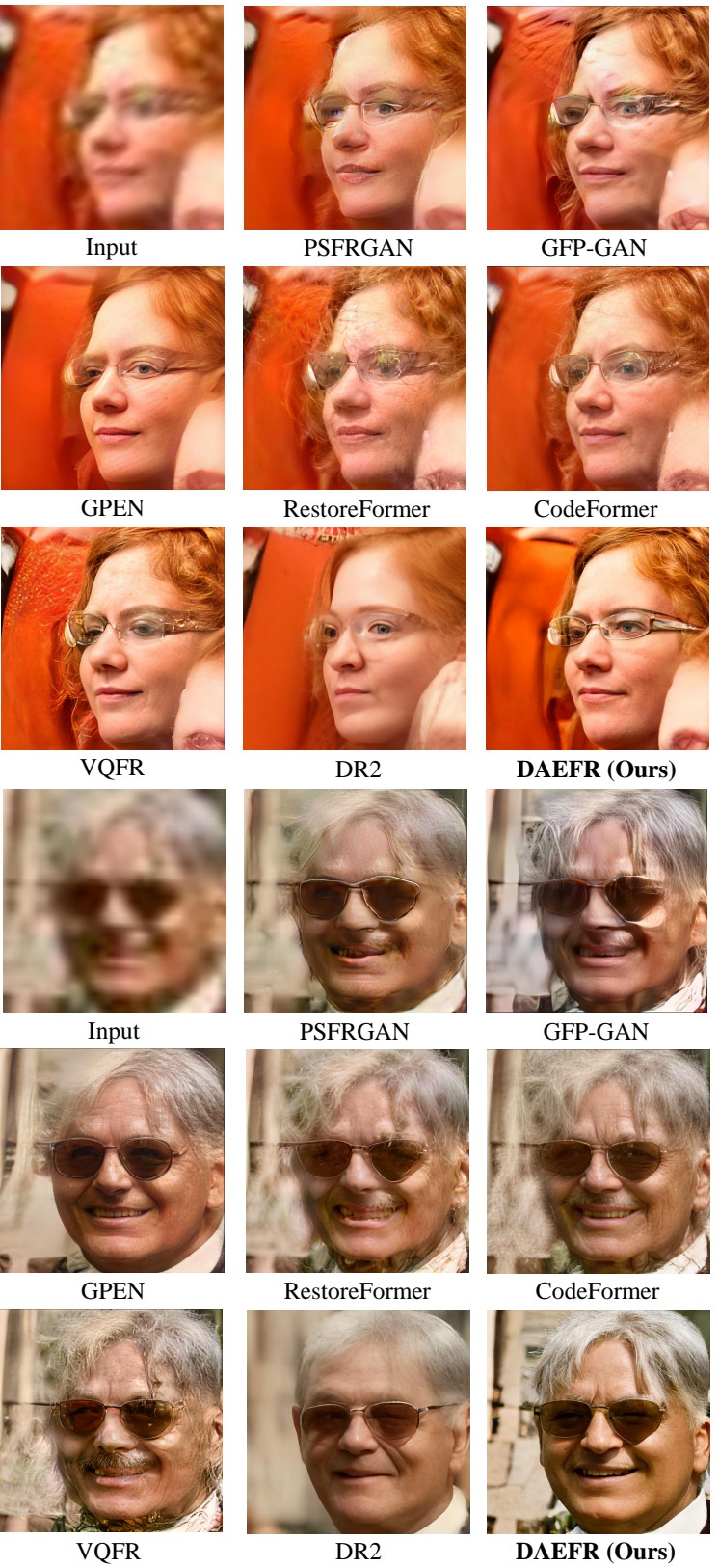

Figure 15: **Qualitative comparison on *WIDER-Test* datasets.** Our DAEFR method exhibits robustness in restoring high-quality faces in detail part.

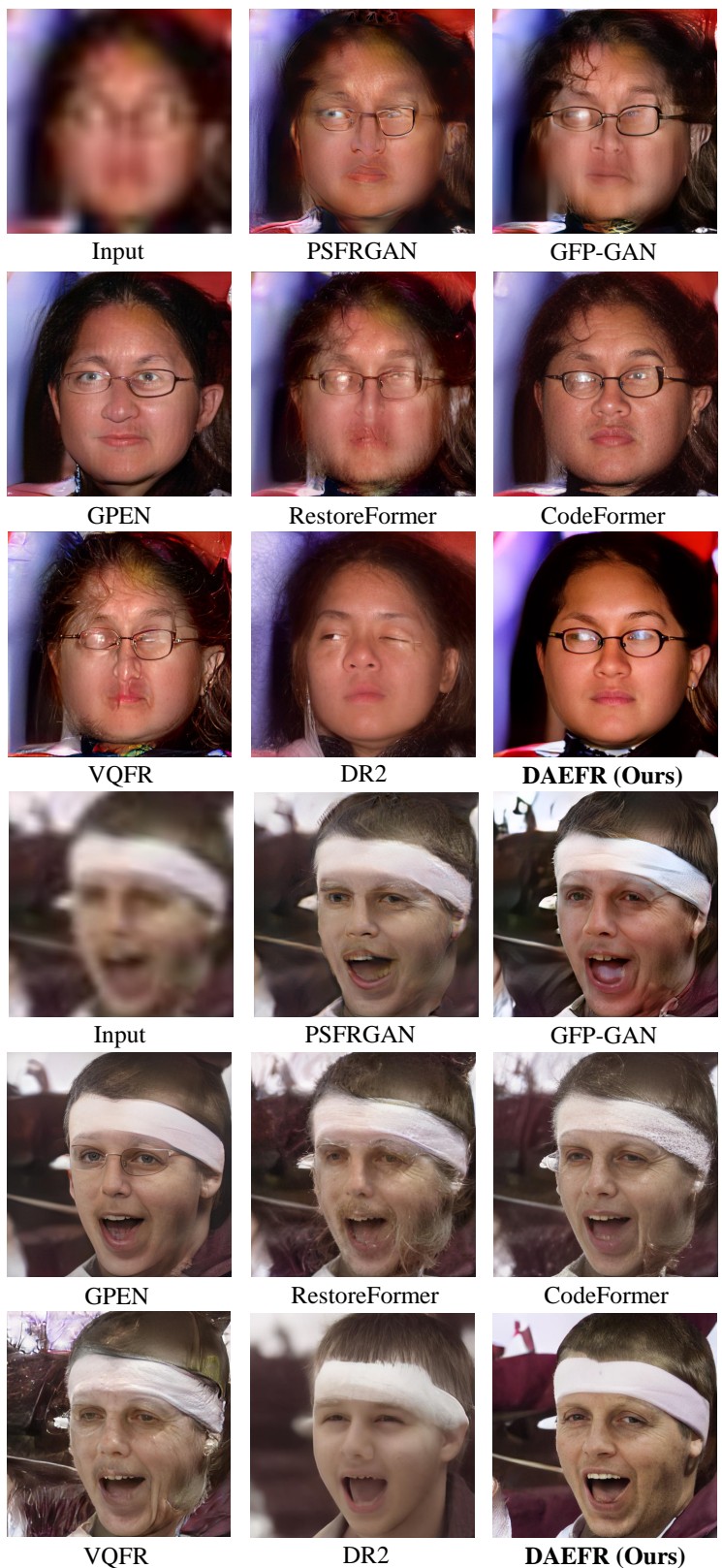

Figure 16: **Qualitative comparison on *WIDER-Test* datasets.** Our DAEFR method exhibits robustness in restoring high-quality faces in detail part.

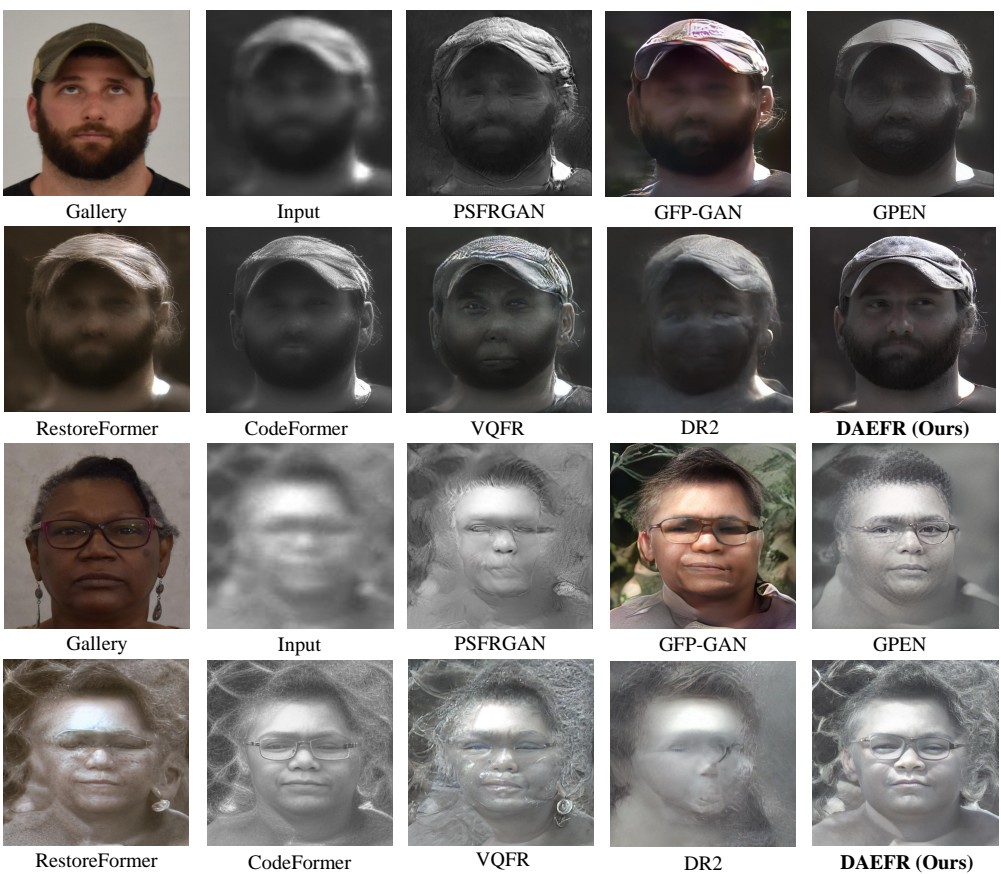

Figure 17: **Qualitative comparison on *BRIAR-Test* datasets.** Our DAEFR method exhibits robustness in restoring high-quality faces in detail part.

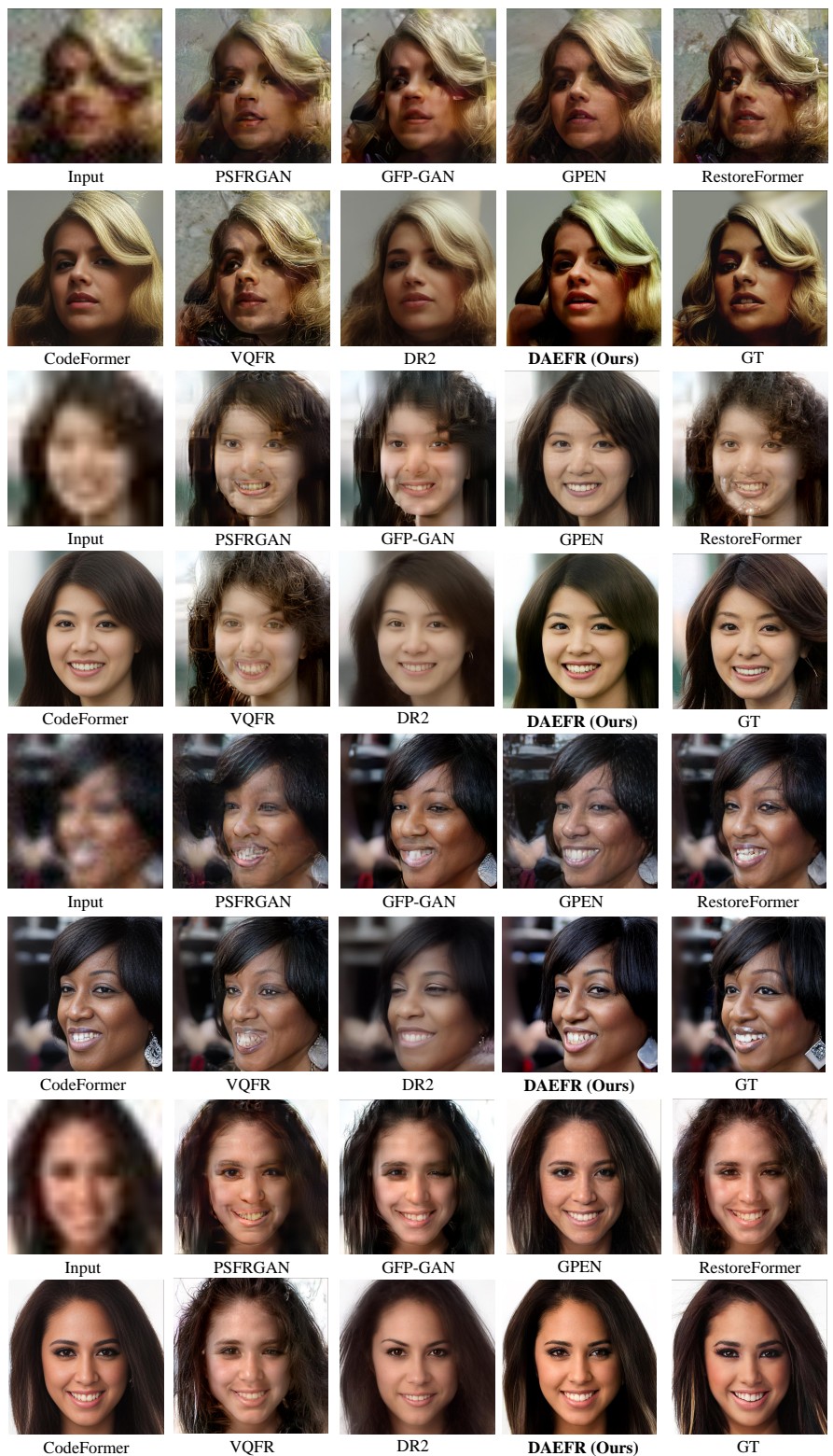

Figure 18: **Qualitative comparison on *CelebA-Test* datasets.** Our DAEFR method exhibits robustness in restoring high-quality faces in detail part.

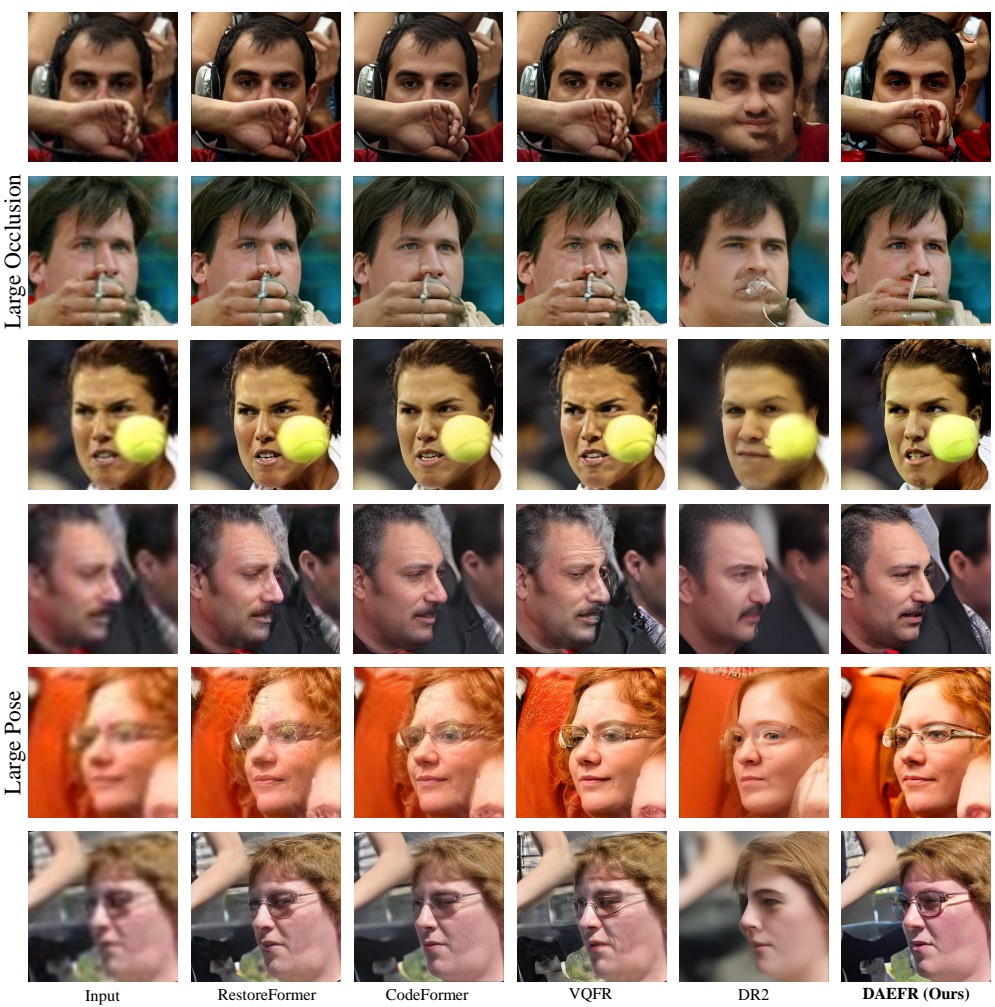

Figure 19: **Qualitative comparison on large occlusion and pose situation.** Our DAEFR method exhibits robustness in restoring high-quality faces in detail part.

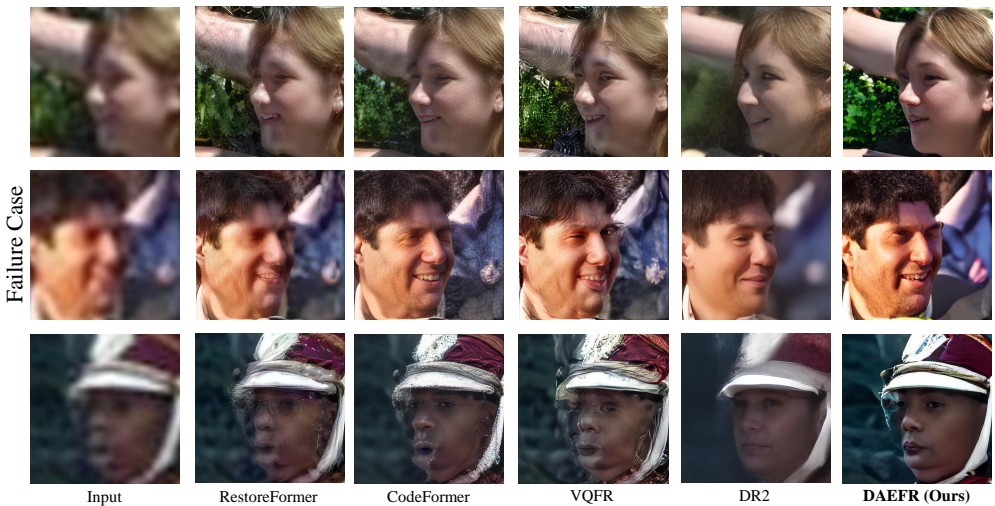

Figure 20: **Failure case on extreme pose situation.** Our DAEFR method fails in extremely pose situations.

