# OpenReview forum: "Dual Associated Encoder for Face Restoration"
_ICLR.cc/2024/Conference — ICLR 2024 poster_

### Official Review · Reviewer_X7AW · 2023-10-31

**Soundness:** 3 good
**Presentation:** 3 good
**Contribution:** 2 fair
**Rating:** 5
**Confidence:** 3

**Summary:**

To address the domain gap between low-quality and high-quality images and improve the performance of face restoration, the paper introduces a novel framework called DAEFR. This framework incorporates LQ (low-quality) image domain information by introducing an auxiliary branch that extracts unique LQ domain-specific features to complement the HQ (high-quality) domain information. To further align the features between the HQ and LQ domains, the paper employs a CLIP-like constraint to enhance the correlation between the two domains. Additionally, to facilitate better feature fusion between these two domains, the framework introduces a multihead cross-attention module. Evaluation results demonstrate the effectiveness of DAEFR.

**Strengths:**

1.	The paper proposes a framework designed to incorporate distinctive features from low-quality (LQ) images, thereby enhancing the face restoration task.
2.	To mitigate the domain gap between HQ and LQ images, the paper proposes an association strategy during training, and incorporates a multihead cross-attention module for better feature fusion between these two domains.
3.	The experiments on both synthetic and real-world datasets demonstrate the effectiveness of the proposed framework.

**Weaknesses:**

1.	The full name of the proposed framework, DAEFR, is missing. It should be mentioned on its first occurrence in the paper.
2.	The proposed method requires training two sets of encoder and decoder for both HQ and LQ images. This will double the training resource requirements.
3.	I think it will be better if there is more elaboration on the domain gap issue that the current works exist, i.e., the motivation of the paper. Currently, it is not intuitive from figure 1 and from current discussion.
4.	Check the spellings. For example, “recently” on the beginning of second paragraph in the “Vector Quantized Codebook Prior” of the related work.
5.	There are some confusions about the training process of the network. In the first stage (section 3.1), you firstly train the two autoencoders of LQ and HQ using the codebook loss. After the first-stage training is complete, you train the two encoders using both the codebook loss and the association loss. Why not combine the two stages into one, or just apply the association loss in stage 2? Besides, in stage 3, you state in the Training Objectives that the MHCA and transformer module are trained in this stage. However, from figure 2(c), the two encoders seem not to be frozen during stage 3.
6.	The results in Table 1 indicate that the proposed method does not significantly outperform other methods, especially for the synthetic CelebA-Test dataset.

**Questions:**

Refer to weaknesses.

---

> ### Author Response · Authors · 2023-11-20
> **Rebuttal by Authors**
>
> We extend our gratitude for the positive and constructive feedback.
>
> We would like to address the raised concerns as follows:
>
> ---
> **[Q1] The full name of the proposed framework, DAEFR, is missing. It should be mentioned on its first occurrence in the paper.**
>
> The acronym 'DAEFR' stands for the title of our paper, which is an initialism of "**D**ual **A**ssociated **E**ncoder for **F**ace **R**estoration."
> We will revise these parts in our updated manuscript.
>
> ---
> **[Q2] The proposed method requires training two sets of encoder and decoder for both HQ and LQ images. This will double the training resource requirements.**
>
> In our implementation, obtaining the codebooks, which demand the most training resources, involves training the HQ path for 300 epochs and the LQ path for 200 epochs. In comparison, CodeFormer requires 700 epochs to create its HQ codebook. Despite this, our use of training resources remains less intensive than that of CodeFormer.
>
> ---
> **[Q3] I think it will be better if there is more elaboration on the domain gap issue that the current works exist, i.e., the motivation of the paper. Currently, it is not intuitive from Figure 1 and the current discussion.**
>
> * We thoroughly analyze the domain gap in three distinct pathways: HQ reconstruction path, LQ reconstruction path, and Hybrid path. All three ways involve the utilization of LQ images as input. As Figure 1 shows in our supplementary material, both the HQ and Hybrid paths exhibit limitations in effectively reconstructing or restoring the LQ image due to the domain gap. Notably, the Hybrid path fails to generate any facial features.
> * In these experiments, we observe that the LQ path better reconstructs the identity information from the LQ images, and the domain gap actually exists. Based on these observations, we design the additional LQ path to better encode the domain-specific information and introduction association stage and feature fusion to solve the domain gap issue.
>
> ---
> **[Q4] Check the spelling. For example, "recently" at the beginning of the second paragraph in the "Vector Quantized Codebook Prior" of the related work.**
>
> We will revise these parts in our updated manuscript.
>
> ---
> **[Q5] Why not combine the two stages or apply the association loss in stage 2? Besides, in stage 3, you state in the Training Objectives that the MHCA and transformer module are trained in this stage. However, from Figure 2(c), the two encoders seem not to be frozen during stage 3.**
>
> * We would like to clarify the role of each stage. In the codebook learning stage, we aim to obtain a pure HQ decoder from the HQ path for use in the final stage. If we are to simply combine the codebook learning stage with the association stage, there is a risk that the HQ decoder might interfere with the LQ path, leading to a decrease in image quality.
> * We also conduct experiments where we apply only the association loss in stage 2. This resulted in less constraint on both paths, potentially causing a misalignment between the HQ and LQ paths. The detailed results are presented in the supplementary material.
> * To address your question, we do not freeze the two encoders in stage 3. We believe that fine-tuning these encoders can accelerate the convergence and enhance feature fusion and code prediction in our restoration process.
>
> ---
> **[Q6] The results in Table 1 indicate that the proposed method does not significantly outperform other methods, especially for the synthetic CelebA-Test dataset.**
>
> In Table 1 of the manuscript, our approach outperforms the state-of-the-art methods, especially when dealing with **real-world datasets**. When it comes to the synthetic CelebA-test dataset, we achieve the **best** performance when we combine the final feature with the LQ features before passing them to the HQ decoder, as shown in the following table. These results highlight the strength and reliability of our approach.
>
> | CelebA-Test       | LPIPS &darr; | PSNR   &uarr; | SSIM &uarr; |
> | ----------------- | ------------ | ------------- | ----------- |
> | RestoreFormer [1] | 0.467        | 20.146        | 0.494       |
> | CodeFormer [2]    | 0.365        | 21.449        | 0.575       |
> | VQFR [3]          | 0.456        | 19.484        | 0.472       |
> | DR2 [4]           | 0.409        | 20.327        | 0.595       |
> | DAEFR (Ours)      | **0.363**    | **21.488**    | **0.616**   |
>
> [1] RestoreFormer: High-Quality Blind Face Restoration from Undegraded Key-Value Pairs. CVPR 2022.
>
> [2] Towards Robust Blind Face Restoration with Codebook Lookup Transformer. NeurIPS 2022.
>
> [3] VQFR: Blind Face Restoration with Vector-Quantized Dictionary and Parallel Decoder. ECCV 2022.
>
> [4] DR2: Diffusion-based Robust Degradation Remover for Blind Face Restoration. CVPR 2023.

---

> > ### Author Response · Authors · 2023-11-22
> > **Please let us know if you have additional questions after reading our response**
> >
> > We appreciate your reviews and comments. We hope our responses address your concerns. Please let us know if you have further questions after reading our rebuttal.
> >
> > We hope to address all the potential issues during the discussion period.
> >
> > Thank you

---

### Official Review · Reviewer_cKwn · 2023-11-01

**Soundness:** 4 excellent
**Presentation:** 4 excellent
**Contribution:** 2 fair
**Rating:** 5
**Confidence:** 4

**Summary:**

In this paper, the authors propose a framework, named dual associated encoder for face restoration (DAEFR), for face restoration. Specifically, different from the existing codebook based methods using only one autoencoder for high-resolution images, the authors propose to add another stream for low-resolution images. To fuse and align the features from both low and high resolution images, an association stage is designed. The associated features then will be extracted and utilized for face restoration.

Experimental results have demonstrated the effectiveness of the proposed method.

**Strengths:**

1. The paper is well written.
2. The idea is well presented, explained, and demonstrated.
3. The proposed method may inspire the researchers in this area.

**Weaknesses:**

1. The contribution looks marginal to me since all the methods used in different stage are well designed and demonstrated. Adding another stream for low-resolution might not be a major contribution for a top-tier venue like ICLR.
2. I got some questions for the experimental results which can be seen in the questions part.

**Questions:**

In Table 2, it seems like all the alter methods outperform the proposed method in terms of LPIPS. Please give discussions or visualizations to explain why this happens.

---

> ### Author Response · Authors · 2023-11-20
> **Rebuttal by Authors**
>
> We extend our gratitude for the positive and constructive feedback.
>
> We would like to address the raised concerns as follows:
>
> ---
> **[Q1] The contribution looks marginal to me since all the methods used in the different stages are well-designed and demonstrated. Adding another stream for low-resolution might not be a major contribution for a top-tier venue like ICLR.**
>
> We would like to further clarify the novelty and effectiveness of this work by putting this work in proper context and emphasizing the empirical results over state-of-the-art approaches.
>
> * We would like to clarify that although some proposed model components are similar to prior arts, it requires **meticulous algorithmic design** and **new modules** to integrate them and achieve state-of-the-art results. As shown in Table 2 of the manuscript, the **straightforward combination** of HQ and LQ features **does not improve performance**. Thus, we introduce the **association stage** and the **feature fusion** modules to exploit the information from both domains effectively. The performance of the proposed model **significantly outperforms** state-of-the-art approaches, as shown in the following table.
>
>
>
> | Ablation Study           | LMD &darr; | NIQE   &darr; |
> | ------------------------ | ---------- | ------------- |
> | Naively combine features | 4.258      | 4.297         |
> | Association & Fusion     | **4.019**  | **3.815**     |
>
>
>
> * We propose an **additional LQ reconstruction path** to specifically encode the LQ domain information, which **does not exist in prior works**. Furthermore, we provide a **new perspective** to handle the LQ domain information better. We show the **effectiveness of our additional LQ path** with extensive experiments. How to better deal with the LQ input, the only source of information in this task, to assist the restoration process is crucial. Our contributions in **providing new encoding strategies and viewpoints to handle LQ information** are essential to advance the field.
>
> * We demonstrate the novel components of our work by comparing with RestoreFormer [1] and CodeFormer [2], whose network building blocks mostly come from VQGAN [3] and Transformer [4]. Although these models are mostly built upon prior works, they show great empirical results can be achieved by **novel and essential integration**. Such works are important and make significant contributions. We use these examples to emphasize that **sufficient novel components** are introduced in proposing an additional LQ encoder to specifically deal with the LQ information and demonstrate its **effectiveness** with extensive experiments. We hope reviewers can appreciate the contribution of this work from this perspective.
>
> * We show the ability of our method in real-world datasets with **significant improvements** over SOTA methods in both quantitative and qualitative evaluation. For the qualitative results, please refer to our manuscript and supplementary material. For the quantitative results, please refer to the following tables.
>
> | LFW-Test          | FID &darr; | NIQE   &darr; |
> | ----------------- | ---------- | ------------- |
> | RestoreFormer [1] | 48.412     | 4.168         |
> | CodeFormer [2]    | 52.350     | 4.482         |
> | VQFR [5]          | 50.712     | 3.589         |
> | DAEFR (Ours)      | **47.532** | **3.552**     |
>
> | WIDER-Test        | FID &darr; | NIQE   &darr; |
> | ----------------- | ---------- | ------------- |
> | RestoreFormer [1] | 49.839     | 3.894         |
> | CodeFormer [2]    | 38.798     | 4.164         |
> | VQFR [5]          | 44.158     | **3.054**     |
> | DAEFR (Ours)      | **36.720** | 3.655         |
>
> [1] RestoreFormer: High-Quality Blind Face Restoration from Undegraded Key-Value Pairs. CVPR 2022.
>
> [2] Towards Robust Blind Face Restoration with Codebook Lookup Transformer. NeurIPS 2022.
>
> [3] Taming Transformers for High-Resolution Image Synthesis. CVPR 2021.
>
> [4] Attention Is All You Need. NeurIPS 2017.
>
> [5] VQFR: Blind Face Restoration with Vector-Quantized Dictionary and Parallel Decoder. ECCV 2022.
>
> ---
> **[Q2] In Table 2, it seems like all the altered methods outperform the proposed method in terms of LPIPS. Please give discussions or visualizations to explain why this happens.**
>
>
> The LPIPS calculation involves using deep neural networks to analyze image features and then calculating a distance or similarity score based on these features. In our case, we input entire images, including background elements that are not our primary restoration focus. These background elements can affect the LPIPS score.
>
> In this ablation study, our main goal is to demonstrate significant improvements in identity metrics. Additionally, we present qualitative results in Fig. 5 of the manuscript, showcasing the best visual outcomes achieved with our final setting.

---

> > ### Author Response · Authors · 2023-11-22
> > **Please let us know if you have additional questions after reading our response**
> >
> > We appreciate your reviews and comments. We hope our responses address your concerns. Please let us know if you have further questions after reading our rebuttal.
> >
> > We hope to address all the potential issues during the discussion period.
> >
> > Thank you

---

### Official Review · Reviewer_qW23 · 2023-11-02

**Soundness:** 3 good
**Presentation:** 3 good
**Contribution:** 3 good
**Rating:** 8
**Confidence:** 3

**Summary:**

This paper introduces a dual-branch framework, named DAEFR, designed for the restoration of high-quality (HQ) facial details from low-quality (LQ) images. Within this framework, an auxiliary LQ encoder and an HQ encoder are employed in conjunction with feature association techniques to capture visual characteristics from LQ images. Subsequently, the features extracted from both encoders are combined to enhance their quality. Finally, the HQ decoder is utilized for the reconstruction of high-quality images. The effectiveness of DAEFR is evaluated using both real-world and synthetic datasets.

**Strengths:**

1. The notion of incorporating an additional encoder with weight sharing is intriguing.

2. The authors have extensively verified the significance of each component via thorough ablation studies.

3. This approach adeptly addresses various common and severe degradations and maintains a high standard of writing quality.

**Weaknesses:**

1. Can you provide a detailed explanation of the primary differentiation between DAEFR and CodeFormer?

2. The paper does not delve into its limitations or potential factors for analysis, which would greatly enrich its discussion.

3. The paper outperforms baseline methods in the downstream face recognition task. Could you provide a comprehensive explanation of these results?

4. The paper does not provide any suggestions or insights into potential avenues for future research or improvements to the proposed method.

**Questions:**

Please discuss the concerns in the Weaknesses Section.

---

> ### Author Response · Authors · 2023-11-20
> **Rebuttal by Authors**
>
> We extend our gratitude for the positive and constructive feedback.
>
> We would like to address the raised concerns as follows:
>
> ---
> **[Q1] Can you provide a detailed explanation of the primary differentiation between DAEFR and CodeFormer?**
>
> * Our method, DAEFR, introduces a crucial distinction by incorporating an additional encoder specifically designed for improved feature encoding in the LQ domain. This allows us to leverage essential information from the LQ images effectively. Moreover, DAEFR uses an association stage that effectively bridges the domain gap between the HQ and LQ domains.
> * In contrast, CodeFormer adopts a different approach by utilizing a pretrained HQ encoder, which is then fine-tuned on LQ images. However, the domain gap still exists despite this adaptation, which may present certain challenges.
> * By integrating the LQ encoder in DAEFR, we aim to encode LQ features better, thereby facilitating more accurate and refined results. This emphasis on enhancing the LQ encoding process demonstrates the critical distinction between DAEFR and CodeFormer, ultimately leading to improved performance and overcoming potential domain gap issues.
>
> ---
> **[Q2] The paper does not delve into its limitations or potential factors for analysis, which would greatly enrich its discussion.**
>
> We have provided visual results with large poses in Figure 20 of the supplementary material. While our method demonstrates robustness in most severe degradation scenarios, we also observe instances where it may fail, particularly in cases with large face poses. This can be expected as the FFHQ dataset contains few samples with large face poses, leading to a scarcity of relevant codebook features to effectively address such situations, resulting in less satisfactory restoration and reconstruction outcomes.
>
> We include failure cases in the supplementary material to ensure a more comprehensive understanding of our method's limitations.
>
> ---
> **[Q3] The paper outperforms baseline methods in the downstream face recognition task. Could you provide a comprehensive explanation of these results?**
>
> Our quantitative experiments demonstrate the strength of our method. The issue with identity evaluation on the synthetic CelebA-test dataset lies in the impracticality of achieving identical images in real-world scenarios. It only measures the likeness between the original image and the restored one. Real face recognition datasets comprise a vast number of images belonging to the same individual, presenting more significant challenges and closely mimicking real-world scenarios. These results confirm the effectiveness of our added LQ encoder, which extracts valuable details from the LQ domain, thereby significantly enhancing the restoration process.
>
> ---
> **[Q4] The paper does not provide any suggestions or insights into potential avenues for future research or improvements to the proposed method.**
>
> We would like to provide detailed suggestions and directions for future research:
>
> * Codebook of Local Facial Parts: As part of future work, we plan to integrate a codebook of local facial parts. This addition could enhance the accuracy of facial restoration, particularly for detailed regions such as the eyes, nose, and mouth, which are crucial for achieving realistic results.
> * Extension to Severe Degraded Video Input: To broaden the applicability of our method, extending it to handle video input is an intriguing direction for future exploration. However, this line of work should consider the consistency of restored results across different time frames. Degradation in videos may vary, and the model should generate coherent and consistent outputs throughout the entire video sequence.

---

> > ### Author Response · Authors · 2023-11-22
> > **Please let us know if you have additional questions after reading our response**
> >
> > We appreciate your reviews and comments. We hope our responses address your concerns. Please let us know if you have further questions after reading our rebuttal.
> >
> > We hope to address all the potential issues during the discussion period.
> >
> > Thank you

---

### Official Review · Reviewer_huYM · 2023-11-02

**Soundness:** 3 good
**Presentation:** 3 good
**Contribution:** 3 good
**Rating:** 8
**Confidence:** 4

**Summary:**

This paper presents a new approach called the Dual Associated Encoder for facial restoration. In this method, an auxiliary Low-Quality (LQ) branch is introduced to extract vital information from LQ inputs. Subsequently, it employs a structure similar to CLIP to establish connections between the LQ and High-Quality (HQ) encoders. This connection aims to reduce the domain gap and information loss when restoring HQ images from LQ inputs. The experimental outcomes illustrate the highly promising performance of this novel approach.

**Strengths:**

1.	The paper offers a coherent and well-founded justification for the research, with a method design that closely aligns with the research objectives.
2.	The paper effectively communicates the method, ensuring readers can easily comprehend the underlying concepts and techniques.
3.	The experimental results showcase remarkable performance, affirming the method's efficacy in tackling the face restoration challenge.

**Weaknesses:**

1.	Absence of Future Research Guidance: The paper does not offer any recommendations or insights into potential future research directions or enhancements for the proposed method.
2.	Omission of Limitation Discourse: The paper lacks a discussion regarding its limitations and possible factors for analysis.

**Questions:**

1.	While the paper predominantly highlights the advantages of the proposed method, could you offer instances where the method encountered shortcomings or limitations?
2.	Could you elaborate on the key distinction between DAEFR and CodeFormer?
3.	Can you provide further experimental details into the "Effectiveness of Low-Quality Feature from Auxiliary Branch" as examined in your ablation studies?

**Details Of Ethics Concerns:**

Please see my comments above.

---

> ### Author Response · Authors · 2023-11-20
> **Rebuttal by Authors**
>
> We extend our gratitude for the positive and constructive feedback.
>
> We would like to address the raised concerns as follows:
>
> ---
> **[Q1] Absence of Future Research Guidance: The paper does not offer any recommendations or insights into potential future research directions or enhancements for the proposed method.**
>
> We would like to provide detailed suggestions and directions for future research:
>
> * Codebook of Local Facial Parts: As part of future work, we plan to integrate a codebook of local facial parts. This addition could enhance the accuracy of facial restoration, particularly for detailed regions such as the eyes, nose, and mouth, which are crucial for achieving realistic results.
> * Extension to Severe Degraded Video Input: To broaden the applicability of our method, extending it to handle video input is an intriguing direction for future exploration. However, this line of work should consider the consistency of restored results across different time frames. Degradation in videos may vary, and the model should generate coherent and consistent outputs throughout the entire video sequence.
>
> ---
> **[Q2] Omission of Limitation Discourse: The paper lacks a discussion regarding its limitations and possible factors for analysis. Could you offer instances where the method encountered shortcomings or limitations?**
>
> We have provided visual results with large poses in Figure 20 of the supplementary material. While our method demonstrates robustness in most severe degradation scenarios, we also observe instances where it may fail, particularly in cases with large face poses. This can be expected as the FFHQ dataset contains few samples with large face poses, leading to a scarcity of relevant codebook features to effectively address such situations, resulting in less satisfactory restoration and reconstruction outcomes.
>
> We include failure cases in the supplementary material to ensure a more comprehensive understanding of our method's limitations.
>
> ---
> **[Q3] Could you elaborate on the key distinction between DAEFR and CodeFormer?**
>
> * Our method, DAEFR, introduces a crucial distinction by incorporating an additional encoder specifically designed for improved feature encoding in the LQ domain. This allows us to leverage essential information from the LQ images effectively. Moreover, DAEFR uses an association stage that effectively bridges the domain gap between the HQ and LQ domains.
> * In contrast, CodeFormer adopts a different approach by utilizing a pretrained HQ encoder, which is then fine-tuned on LQ images. However, the domain gap still exists despite this adaptation, which may present certain challenges.
> * By integrating the LQ encoder in DAEFR, we aim to encode LQ features better, thereby facilitating more accurate and refined results. This emphasis on enhancing the LQ encoding process demonstrates the critical distinction between DAEFR and CodeFormer, ultimately leading to improved performance and overcoming potential domain gap issues.
>
> ---
> **[Q4] Can you provide further experimental details into the "Effectiveness of Low-Quality Feature from Auxiliary Branch" as examined in your ablation studies?**
>
> To demonstrate the effectiveness of our auxiliary LQ branch, we conduct validated experiments of fusing LQ features with feature $Z^{c}\_{f}$ in the feature fusion and code prediction stage. These experiments involve extracting LQ features $Z^{c}\_{l}$ from the LQ codebook and adding a control module [1] [2]. Given a feature control module and feature scalar $s\_{lq}$, we can control the scale of the LQ feature $Z^{c}\_{l}$ to fuse with the feature $Z^{c}\_{f}$ before feeding to the HQ decoder. The network architecture is shown in Figure 10 in the supplementary material.
>
> We conduct an additional training stage to enable the control module to fuse different features effectively. In our implementation, we fine-tune the control module for ten epochs and maintain a feature scalar value of $s\_{lq} =1$ during training.
>
> [1] Recovering realistic texture in image super-resolution by deep spatial feature transform, CVPR 2018.
>
> [2] Towards Robust Blind Face Restoration with Codebook Lookup Transformer. NeurIPS 2022.

---

> > ### Author Response · Authors · 2023-11-22
> > **Please let us know if you have additional questions after reading our response**
> >
> > We appreciate your reviews and comments. We hope our responses address your concerns. Please let us know if you have further questions after reading our rebuttal.
> >
> > We hope to address all the potential issues during the discussion period.
> >
> > Thank you

---

### Official Review · Reviewer_UndL · 2023-11-30

**Soundness:** 3 good
**Presentation:** 3 good
**Contribution:** 3 good
**Rating:** 8
**Confidence:** 4

**Summary:**

The paper presents "DAEFR," a novel framework for blind face restoration, specifically focusing on restoring high-quality facial images from low-quality ones. This challenge arises due to complex and unknown sources of degradation in facial images. The key innovation in DAEFR is the introduction of an auxiliary low-quality (LQ) encoder, trained exclusively on LQ data. This branch captures domain-specific information from LQ inputs, addressing the domain gap and distinct feature representations between LQ and HQ images.

DAEFR's methodology involves discrete codebook learning for both HQ and LQ domains, feature association between the HQ and LQ encoders, and a feature fusion stage that combines the information from both encoders. This approach is designed to overcome limitations in existing codebook methods, which often exhibit domain bias due to a reliance on encoders pre-trained solely on HQ data.

The paper's contributions are significant in several aspects:
*  Introduction of the auxiliary LQ encoder for more accurate LQ domain feature representation.
* Utilization of association and feature fusion methods to effectively bridge the domain gap between LQ and HQ images, enhancing restoration outcomes.
* A comprehensive evaluation of DAEFR on synthetic and real-world datasets demonstrates superior performance in restoring facial details compared to existing state-of-the-art methods.
Overall, the paper proposes a novel and effective approach to address the challenges of blind face restoration under severe degradation, emphasizing maintaining the fidelity and identity information present in the original LQ images.

**Strengths:**

The paper's strengths are notable across various dimensions, including originality, quality, clarity, and significance:

### Originality
- **Innovative Approach**: Introducing an auxiliary LQ encoder specifically trained on LQ data is a creative solution to the domain gap problem in image restoration. This approach significantly differs from conventional methods, primarily relying on encoders pre-trained on HQ data.
- **Feature Fusion and Association Techniques**: The application of feature association techniques and the subsequent fusion of features from both LQ and HQ encoders demonstrate a novel integration of ideas, enhancing the restoration process's effectiveness.

### Quality
- **Comprehensive Evaluation**: The method is thoroughly evaluated on synthetic and real-world datasets, providing substantial evidence of its effectiveness. The comparison with state-of-the-art methods further underlines the quality of the proposed approach.
- **Robustness and Fidelity**: The paper demonstrates the robustness of the DAEFR method in preserving the identity and details in restored images, even under severe degradation conditions.

### Clarity
- **Well-Structured Presentation**: The paper is well-organized, with each section logically flowing into the next. The methodology, experimental setup, and results are clearly explained, making them accessible to readers.
- **Effective Use of Visual Aids**: Including figures and tables aids in illustrating the methodology and showcasing the results, enhancing the overall clarity of the paper.

### Significance
- **Contribution to Blind Face Restoration**: The paper addresses a significant challenge in image restoration, particularly in restoring HQ images from LQ ones. The proposed solution can potentially influence future research directions in this area.
- **Applicability and Impact**: The approach can greatly interest researchers and practitioners in computer vision, offering a novel tool for addressing a common problem in image restoration. Its applicability to real-world, severely degraded images enhances its significance.

**Weaknesses:**

The paper, while strong in many aspects, does have some areas that could be improved upon for a more comprehensive understanding and assessment of the proposed method:

1. **Generalization to Other Domains**: The paper focuses on facial image restoration. It would be beneficial to see how the proposed DAEFR method performs in other domains of image restoration or different types of image degradation beyond facial images. This expansion could provide insights into the method's versatility and applicability in broader contexts.

2. **Comparison with Varied Methods**: While the paper compares DAEFR with state-of-the-art methods, it primarily focuses on similar restoration approaches. Including comparisons with diverse methods, especially those using different underlying principles or restoration techniques could provide a more rounded evaluation of DAEFR's performance.

3. **Limitation in Handling Extremely Diverse Degradations**: The paper could explore the limitations of the DAEFR method in handling extremely diverse or uncommon types of image degradations. Understanding these limitations would be critical for practical applications and could guide future improvements to the method.

4. **Computational Efficiency and Scalability**: The paper does not delve deeply into the computational efficiency or scalability of the proposed method. Details on the computational resources required, processing time, and scalability to larger datasets or higher-resolution images would be valuable for assessing the practicality of DAEFR in real-world scenarios.

5. **Detailed Analysis of Auxiliary Encoder's Impact**: While introducing an auxiliary LQ encoder is a key feature of the paper, a more detailed analysis of how this encoder improves the restoration process would be insightful. This could include comparisons between results with and without the auxiliary encoder or exploring its impact on different types of degradation.

6. **User Study for Qualitative Assessment**: Incorporating a user study to supplement the quantitative evaluations could provide additional insights into the perceptual quality of the restored images. This would be particularly useful in understanding the real-world applicability of the method.

Addressing these aspects could strengthen the paper's contributions and provide a more comprehensive understanding of the DAEFR method's capabilities and potential areas for further development.

**Questions:**

1. **Generalization to Other Image Types**: How well does the DAEFR method generalize to other images beyond facial images? Are there specific types of image degradation or different domains where DAEFR might not perform as effectively?

2. **Handling Extreme Degradations**: Could you provide more insights into how DAEFR performs under extremely diverse or uncommon image degradations? Understanding its limitations in such scenarios would be crucial for practical applications.

3. **Computational Efficiency**: Can you provide details regarding the computational efficiency of DAEFR, such as processing time and resource requirements? How scalable is the method for larger datasets or higher-resolution images?

4. **Impact of Auxiliary Encoder**: Could you elaborate on the specific impact of the auxiliary LQ encoder in the restoration process? For instance, how does the restoration quality differ when the auxiliary encoder is not used?

5. **Comparison with Diverse Restoration Methods**: The paper compares DAEFR with similar restoration approaches. Could you compare its performance with restoration techniques that use fundamentally different principles?

6. **User Study for Qualitative Assessment**: Have you considered conducting a user study to evaluate the perceptual quality of the restored images? This could provide valuable insights into the real-world applicability of DAEFR.

7. **Further Methodological Details**: Could you provide more technical details or insights on the feature association techniques used? How do they specifically contribute to bridging the domain gap between LQ and HQ images?

Responses to these questions could greatly enhance the understanding of DAEFR's capabilities, limitations, and potential areas for future development.

---

### Comment · Area_Chair_BEnR · 2023-11-23
**[ICLR 2024 Reviewers’ feedback] Please read authors’ responses and give your feedback**

Dear Reviewers,

Thanks again for your strong support and contribution as an ICLR 2024 reviewer.

Please check the response and other reviewers’ comments. You are encouraged to give authors your feedback after reading their responses. Thanks again for your help!

Best,

AC

---

### Meta-Review · Area_Chair_BEnR · 2023-12-12

**Metareview:**

The authors propose a dual associated encoder for face restoration. They also propose a cross-attention module for better feature fusion between low-quality and high-quality domains. Experimental results have demonstrated the effectiveness of the proposed method.

Most reviewers give high scores. Only one reviewer gave a negative score.

**Justification For Why Not Higher Score:**

There are still some details raised by the reviewers to be further improved.

**Justification For Why Not Lower Score:**

Most reviewers give high scores. The authors's rebuttal addresses them well.

---

### Decision · Program_Chairs · 2024-01-16

Accept (poster)